# Simultaneous confidence intervals for all pairwise comparisons of the means of delta-lognormal distributions with application to rainfall data

**Patcharee Maneerat[1], Sa-Aat Niwitpong[2], Suparat Niwitpong**[2]*

**1** Department of Mathematics, Uttaradit Rajabhat University, Uttaradit, Thailand, **2** Department of Applied Statistics, King Mongkut's University of Technology North Bangkok, Bangkok, Thailand

* suparat.n@sci.kmutnb.ac.th

**Data Availability Statement:** All relevant data are within the paper and its Supporting information files.

## Abstract

Natural disasters such as flooding and landslides are important unexpected events during the rainy season in Thailand, and how to direct action to avoid their impacts is the motivation behind this study. The differences between the means of natural rainfall datasets in different areas can be estimated using simultaneous confidence intervals (SCIs) for pairwise comparisons of the means of delta-lognormal distributions. Our proposed methods are based on a parametric bootstrap (PB), a fiducial generalized confidence interval (FGCI), the method of variance estimates recovery (MOVER), and Bayesian credible intervals based on mixed (BCI-M) and uniform (BCI-U) priors. Their coverage probabilities, lower and upper error probabilities, and relative average lengths were used to evaluate and compare their SCI performances through Monte Carlo simulation. The results show that BCI-U and PB work well in different situations, even with large differences in variances $\sigma_j^2$. All of the methods were applied to estimate pairwise differences between the means of natural rainfall data from five areas in Thailand during the rainy season to determine their abilities to predict occurrences of flooding and landslides.

## Introduction

Thailand, which is located above the Equator in the tropical zone, is in the center of Southeast Asia and shares borders with Laos (north and east), Myanmar (north and west), Cambodia (east), and Malaysia (south). Its total area is 513,115 square kilometers [1] and its population is over 60 million. From climatological and meteorological perspectives, Thailand is divided into northern, northeastern, central, eastern, and southern (east and west coast) areas. Past natural disasters in Thailand have involved flooding, drought, tropical storms, earthquakes, landslides, and forest fires. When considering natural rainfall in the rainy season (mid-May to mid-October), heavy storms can cause flooding and landslides. Moreover, basins, caves, and waterfalls are especially susceptible to dangerous flash flooding during the rainy season. In July 2018, two situations in the northern area caused significant loss of life, affected transportation, and

**Funding:** This research was funded by King Mongkut's University of Technology North Bangkok. Grant number: KMUTNB-BasicR-64-32.

**Competing interests:** The authors have declared that no competing interests exist.

damaged property and infrastructure. In a recent well-known incident, flash flooding trapped twelve football players and their coach in the Tham Luang cave complex in northern Thailand that triggered a major international search and rescue effort [2]. In another incident, landslides due to heavy downpours for around two weeks claimed victims from seven hill tribes in the northern region [3].

A delta-lognormal distribution can be applied to data having a mixture of non-negative and zero observations: the zero values are binomially distributed with the proportion of zeros $\delta$, while the positive values follow log-normality with mean $\mu$ and variance $\sigma^2$ on a logarithmic scale. Inference on the delta-lognormal mean has been applied to real-world datasets in several fields, such as climatology [4], fishery survey [5, 6], environmental studies [7, 8], and medical treatment [8, 9]. There are two main methodologies in probability and statistical inference to estimate a parameter of the model: point and interval estimation and hypothesis testing. The most well-known interval estimation technique is the confidence interval (CI), and many studies have focused on CIs for a single delta-lognormal mean [5, 8, 10–13]. Meanwhile, some authors have expanded the concept to the comparison between two delta-lognormal means [4, 14], and especially, others have constructed CIs for the difference between two delta-lognormal means [15, 16]. The simultaneous confidence interval (SCI) can be used to estimate all of the parameters of interest in a model at the same time. Constructing CIs for them makes it necessary to perform multiple comparisons provided that each dataset is independently and identically distributed (i.i.d.). In some situations, the SCI might be more important than the CIs for individual parameters [17]. Moreover, no studies have yet been conducted on constructing SCIs for pairwise comparisons of delta-lognormal means.

The aim of the present study is to estimate the differences between the means of the natural rainfall datasets of five areas in Thailand during the rainy season using constructed SCIs for pairwise comparisons of delta-lognormal means of more than two populations ($k > 2$). Herein, we propose five methods for constructing SCIs: the parametric bootstrap (PB), the fiducial generalized confidence interval (FGCI), the method of variance estimates recovery (MOVER), and Bayesian credible intervals based on mixed (BCI-M) and uniform (BCI-U) priors. The PB is extended from SCIs for the ratio of lognormal means provided by Sadooghi-Alvandi and Malekzadeh [18]. Moreover, we constructed the FGCI based on the fiducial generalized pivotal quantity (FGPQ) of the non-zero proportion ($\delta'$) [8, 13]. Motivated by Donner and Zou [19], we used the Wilson interval for $\delta'$ together with the CI for the lognormal mean to formulate the MOVER-based SCI. The BCIs were developed using the different priors: mixed (BCI-M based on normal and inverse Chi-squared distributions) and uniform (BCI-U based on normal approximation and a beta distribution) motivated by Harvey and Merwe [4] and Maneerat et al. [14], respectively.

Motivated by these studies, our contribution to the field is developing and constructing SCIs based on our proposed methods to elucidate the pairwise differences between the means of multiple delta-lognormal distributions. As a practical demonstration, we estimated the pairwise differences between the means of natural rainfall records from the five areas of Thailand. Importantly, this approach could be used to recognize and predict natural disasters in a particular area. From the Akaike information criterion (AIC) results, histograms, and normal Q-Q plots, the rainfall data from all of the study areas follow delta-lognormal distributions except for the northeastern area, which was thus excluded from the study.

The rest of this paper is organized as follows. The concepts and computational procedures of the proposed methods are elaborated in the materials and methods section. Next, simulation studies and numerical results for sample cases $k = 3, 5$ are presented, followed by computation and interpretation of the estimated differences between the means of the rainfall datasets of the five areas in Thailand. Finally, discussion and conclusions on the study are presented.

## Materials and methods

Let $\mathbf{W} = (W_{j1}, W_{j2}, \ldots, W_{jn_j})$ be independent and identically distributed random vector from an $k$-dimensional delta-lognormal distribution $\Delta(\boldsymbol{\mu}, \boldsymbol{\sigma}^2, \boldsymbol{\delta})$, where $\boldsymbol{\mu} = (\mu_1, \mu_2, \ldots, \mu_k)'$, $\boldsymbol{\sigma}^2 = (\sigma_1^2, \sigma_2^2, \ldots, \sigma_k^2)'$, and $\boldsymbol{\delta} = (\delta_1, \delta_2, \ldots, \delta_k)'$; $j = 1, 2, \ldots, k$. For $W_j > 0$, $\mathbf{Y} = \ln \mathbf{W} = (Y_{j1}, Y_{j2}, \ldots, Y_{jn_{j(1)}}) \sim N(\boldsymbol{\mu}, \boldsymbol{\sigma}^2 I)$, where $n_{j(1)} = \#\{j : W_{jn_j} > 0\}$, $n_j = n_{j(0)} + n_{j(1)}$, and $I$ denotes the identity matrix of dimension $n_{j(1)}$. The number of zeros, $n_{j(0)}$, has a binomial distribution $Bi(n_j, \delta_j)$. For $W_j > 0$, Aitchison and Brown [20] defined the distribution of $\mathbf{W}$ as

$$H(W_j; \mu_j, \sigma_j^2, \delta_j) = \delta_j + (1 - \delta_j) G(W_j; \mu_j, \sigma_j^2)$$

where $G(W_j; \mu_j, \sigma_j^2)$ is a lognormal distribution. For $W_j = 0$, $H(W_j; \mu_j, \sigma_j^2, \delta_j) = \delta_j$. Let $\bar{W}_j = n_{j(1)}^{-1} \sum_{r=1}^{n_{j(1)}} Y_{jr}$ and $S_j^2 = (n_{j(1)} - 1)^{-1} \sum_{r=1}^{n_{j(1)}} (Y_{jr} - \bar{W}_j)^2$ be the sample mean and variance for the log-transformed observations, respectively, and $\hat{\delta}_j = n_{j(0)}/n_j$ be the sample proportion of zero based on the $j^{th}$ sample. Thus, the population mean of $W_j$ is

$$\theta_j = (1 - \delta_j) \exp\left(\mu_j + \sigma_j^2/2\right)$$

Using $\bar{W}_j$, $S_j^2$ and $\hat{\delta}_j$ from the samples, the uniformly minimum variance unbiased (UMVU) estimate of $\theta_j$ can be expressed [20]

$$\hat{\theta}_j = \begin{cases} 0 & ; \ n_{j(1)} = 0 \\ W_{j1}/n_j & ; \ n_{j(1)} = 1 \\ (1 - \hat{\delta}_j) \exp\left(\bar{W}_j\right) \psi_{n_{j(1)}}(S_j^2/2) & ; \ n_{j(1)} > 1 \end{cases}$$

where the Bessel function $\psi_a(b)$ is defined as

$$\psi_a(b) = 1 + \frac{(a - 1)b}{a} + \frac{(a - 1)^3}{a^2 2!} \frac{b^2}{a + 1} + \frac{(a - 1)^5}{a^3 3!} \frac{b^3}{(a + 1)(a + 3)} + \cdots$$

According to Crow and Shimizu [21], the asymptotic expansion in power to $1/n_j$ of the variance of $\hat{\theta}_j$; $n_{j(1)} > 1$ is given by

$$\begin{aligned} Var(\hat{\theta}_j) &= \exp\left(2\mu_j + \sigma_j^2\right) \left[ \frac{1}{n_j^2} \sum_{r=1}^{n_j} \binom{n_j}{r} (1 - \delta_j)^r \delta_j^{n_j - r} r^2 \exp\left(\frac{\sigma_j^2}{r}\right) \right. \\ &\qquad \left. {}_0F_1\left(\frac{r - 1}{2}; \frac{(r - 1)^2}{4r^2} \sigma_j^4\right) - (1 - \delta_j)^2 \right] \\ &= \frac{\exp\left(2\mu_j + \sigma_j^2\right)}{n_j} \left[ \delta_j(1 - \delta_j) + \frac{1}{2}(1 - \delta_j)(2\sigma_j^2 + \sigma_j^4) \right] + O(n_j^{-2}) \end{aligned}$$

(1)

where the hypergeometric function is ${}_0F_1(a; b) = \sum_{m=0}^{\infty} \frac{b^m}{(a)_m m!}$, where

$$(a)_m = \begin{cases} 1 & ; m = 0 \\ a(a + 1) \ldots (a + m - 1) & ; m \geq 1 \end{cases}$$

Since our focus is on all pairwise differences among the means of delta-lognormal

distributions, then

$$\theta_{jl} = \theta_j - \theta_l$$

for $\forall j \neq l$; $j, l = 1, 2, \ldots, k$, for which we can obtain estimates $\hat{\theta}_{jl} = \hat{\theta}_j - \hat{\theta}_l$. Note that since $\mathbf{W} = (W_{j1}, W_{j2}, \ldots, W_{jn_j})$ be random samples from delta-lognormal distribution. In agreement with Crow and Shimizu [21], this leads to obtain the UMVU estimates $\hat{\theta}_j$ and $\hat{\theta}_l$ which have the properties of random variables. Thus, the pairwise covariance between them is $COV(\hat{\theta}_j, \hat{\theta}_l) = 0$. From Eq (1), the variance of $\hat{\theta}_{jl}$ can be written as

$$Var(\hat{\theta}_{jl}) = Var(\hat{\theta}_j) + Var(\hat{\theta}_l)$$

By substituting the estimates $(\bar{W}_j, S_j^2, \hat{\delta}_j)$ and $(\bar{W}_l, S_l^2, \hat{\delta}_l)$ from the samples, the approximation of the estimated variance $Var(\hat{\theta}_{jl})$ is obtained as

$$V_{\hat{\theta}_{jl}} = V_{\hat{\theta}_j} + V_{\hat{\theta}_l}$$

$$\approx \frac{\exp\left(2\bar{W}_j + S_j^2\right)}{n_j}\left[\hat{\delta}_j(1 - \hat{\delta}_j) + \frac{1}{2}(1 - \hat{\delta}_j)(2S_j^2 + S_j^4)\right]$$

$$+ \frac{\exp\left(2\bar{W}_l + S_l^2\right)}{n_l}\left[\hat{\delta}_l(1 - \hat{\delta}_l) + \frac{1}{2}(1 - \hat{\delta}_l)(2S_l^2 + S_l^4)\right]$$

The methods to construct the SCIs for $\theta_{jl}$ *are elaborated as follows.*

## The PB interval

Let $\bar{W}_j^{(obs)}$, $S_j^{2(obs)}$ and $\hat{\delta}_j^{\prime(obs)} = 1 - \hat{\delta}_j^{(obs)}$ be the observed values of $\bar{W}_j$, $S_j^2$ and $\hat{\delta}_j^\prime$; $n_{j(1)}^{(obs)} = n\hat{\delta}_j^{\prime(obs)}$. These can be used to represent the estimated values of parameters $\mu_j$, $\sigma_j^2$, and $\delta_j^\prime$, thereby obtaining the empirical distribution of $T$ based on PB. According to Sadooghi-Alvandi and Malekzadeh [18], it is well-known that $Z_j = \sqrt{n_{j(1)}^{(obs)}}[\bar{W}_j^{(B)} - \bar{W}_j^{(obs)}]/S_j^{(obs)} \sim N(0, 1)$ and $U_j = [n_{j(1)}^{(obs)} - 1]S_j^{2(B)}/S_j^{2(obs)} \sim \chi^2_{n_{j(1)}^{(obs)} - 1}$ are independent random variables, and so

$$\bar{W}_j^{(B)} = Z_j \frac{S_j^{(obs)}}{\sqrt{n_{j(1)}^{(obs)}}} \tag{2}$$

$$S_j^{2(B)} = \frac{S_j^{2(obs)}}{n_{j(1)}^{(obs)} - 1} U_j \tag{3}$$

which are their sampling distributions when the values of the nuisance parameters $\sigma_j^2$ are replaced by $S_j^{2(obs)}$, while the values of $\mu_i$ are fixed at 0. The sampling distribution of $\hat{\delta}_j^\prime$ can be considered as a beta distribution, as motivated by Hasan and Krishnamoorthy [8]:

$$\hat{\delta}_j^{\prime(B)} \sim \text{beta}(n_{j(1)} + 0.5, n_{j(0)} + 0.5) \tag{4}$$

which also satisfies the fiducial generalized pivotal quantity (FGPQ) conditions in the FGCI interval where $n_{j(1)} = n_j\hat{\delta}_j^\prime$ and $n_{j(0)} = n_j - n_{j(1)}$. The PB variable based on the pivotal quantity

can be written as

$$T^{(PB)} = \frac{|\{\hat{\delta}_j'^{(B)} \exp(\bar{W}_j^{(B)}) \psi_{n_{j(1)}^{(B)}}(S_j^{2(B)}/2) - \hat{\delta}_l'^{(B)} \exp(\bar{W}_l^{(B)}) \psi_{n_{l(1)}^{(B)}}(S_l^{2(B)}/2)\} - \hat{\theta}_{jl}^{(obs)}|}{\sqrt{V_{\theta_{jl}}^{(B)}}}$$

where $n_{j(1)}^{(B)} = n_j \hat{\delta}_j'^{(B)}, n_{l(1)}^{(B)} = n_l \hat{\delta}_l'^{(B)}$,

$$\psi_{n_{j(1)}^{(B)}}(S_j^{2(B)}/2) \quad = 1 + \frac{(n_{j(1)}^{(B)} - 1)(S_j^{2(B)}/2)}{n_{j(1)}^{(B)}} + \frac{(n_{j(1)}^{(B)} - 1)^3}{n_{j(1)}^{2(B)} 2!} \frac{(S_j^{2(B)}/2)^2}{n_{j(1)}^{(B)} + 1} + \cdots$$

$$\psi_{n_{l(1)}^{(B)}}(S_l^{2(B)}/2) \quad = 1 + \frac{(n_{l(1)}^{(B)} - 1)(S_l^{2(B)}/2)}{n_{l(1)}^{(B)}} + \frac{(n_{l(1)}^{(B)} - 1)^3}{n_{l(1)}^{2(B)} 2!} \frac{(S_l^{2(B)}/2)^2}{n_{l(1)}^{(B)} + 1} + \cdots$$

$$\hat{\theta}_{jl}^{(obs)} \quad = \hat{\delta}_j'^{(obs)} \exp(\bar{W}_j^{(obs)}) \psi_{n_{j(1)}}(S_j^{2(obs)}) - \hat{\delta}_l'^{(obs)} \exp(\bar{W}_l^{(obs)}) \psi_{n_{l(1)}}(S_l^{2(obs)})$$

and

$$V_{\theta_{jl}}^{(B)} \quad \approx \frac{\exp(2\bar{W}_j^{(B)} + S_j^{2(B)})}{n_j} \left[ \hat{\delta}_j'^{(B)}(1 - \hat{\delta}_j'^{(B)}) + \frac{1}{2} \hat{\delta}_j'^{(B)}(2S_j^{2(B)} + S_j^{4(B)}) \right]$$

$$+ \frac{\exp(2\bar{W}_l^{(B)} + S_l^{2(B)})}{n_l} \left[ \hat{\delta}_l'^{(B)}(1 - \hat{\delta}_l'^{(B)}) + \frac{1}{2} \hat{\delta}_l'^{(B)}(2S_l^{2(B)} + S_l^{4(B)}) \right]$$

Therefore, the $100(1 - \alpha)$%PB-based SCI for $\theta_{jl}$ is written as

$$SCI_{\theta_{jl}}^{(PB)} = [\hat{\theta}_{jl} \mp q_\alpha^{(PB)} \sqrt{V_{\theta_{jl}}}] \tag{5}$$

where $q_\alpha^{(PB)}$ is the $(1 - \alpha)^{th}$ percentile of the distribution of $T^{(PB)}$. The PB-based SCI in Eq (5) has the asymptotic coverage property, as demonstrated in Theorem 1 in S1 Appendix. The proof of Theorem 1 in S1 Appendix is similar to Hanning et al. [22], Kharrati-Kopaei and Eftekhar [23], Li et al. [24]. From Eq 5, we can imply that there are no differences between the SCIs for two or more groups due to the constructed SCIs used in multiple comparisons. This fact is also similar to the formulated MOVER interval.

*Algorithm 1*: *The PB interval*

```
1 Generate W̄ⱼ⁽ᴮ⁾, Sⱼ²⁽ᴮ⁾, and δ̂ⱼ′⁽ᴮ⁾ as given by Eqs (2), (3) and (4),
respectively,
2 Compute T⁽ᴾᴮ⁾.
3 Repeat steps 1-2, a large number of times, m = 2500. The empirical
distribution of T⁽ᴾᴮ⁾ is obtained to compute qα⁽ᴾᴮ⁾.
4 Compute 95%SCI-based PB for θⱼₗ, as given by Eq (5).
```

## The FGCI interval

Hannig et al. [25] claim that a variant of the FGPQ1 and FGPQ2 conditions, is stronger than the GPQ2 condition of Weerahandi [26]. Hence, we developed our FGCI to establish SCIs for $\theta_{jl}$. Let $(\bar{W}_j, \bar{W}_j^*)$, $(S_j^2, S_j^{2*})$ and $(\hat{\delta}_j, \hat{\delta}_j^*)$ be i.i.d. random variables such that $\bar{W}_j^*$, $S_j^{2*}$, and $\hat{\delta}_j^*$ are

independent copies of $\bar{W}_j$, $S_j^2$, and $\hat{\delta}_j$. The FGPQs of $\mu_j$ and $\sigma_j^2$ are given by [27]

$$
\begin{aligned}
(R_{\mu_j}, R_{\sigma_j^2}) &= \left( \bar{W}_j - T_j \sqrt{\frac{(n_{j(1)} - 1)}{U_j} \frac{S_j^2}{n_{j(1)}}}, \frac{(n_{j(1)} - 1)}{U_j} S_j^2 \right) \\
&= \left( \bar{W}_j - \frac{[\bar{W}_j^* - \mu_j]}{\sqrt{\sigma_j^2/n_{j(1)}}} \sqrt{\frac{\sigma_j^2}{S_j^{2*}} \frac{S_j^2}{n_{j(1)}}}, \frac{\sigma_j^2}{S_j^{2*}} S_j^2 \right)
\end{aligned}
\tag{6}
$$

Hasan and Krishnamoorthy [8] developed the FGPQ of $\delta_j' = 1 - \delta_j$ as

$$
R_{\delta_j'} \sim \text{beta}(n_{j(1)} + 0.5, n_{j(0)} + 0.5)
\tag{7}
$$

From Eqs (6) and (7), we can obtain

$$
R_{\theta_{jl}}(\boldsymbol{Y}, \boldsymbol{Y}^*, \boldsymbol{\mu}, \boldsymbol{\sigma^2}, \boldsymbol{\delta}) = R_{\delta_j'} \exp\left(R_{\mu_j} + R_{\sigma_j^2}/2\right) - R_{\delta_l'} \exp\left(R_{\mu_l} + R_{\sigma_l^2}/2\right)
\tag{8}
$$

Therefore, the simultaneous $100(1 - \alpha)\%$ FGCI for $\theta_{jl}$ is given by

$$
SCI_{\theta_{jl}}^{(FGCI)} = [\hat{\theta}_{jl} \mp t_\alpha^{(FGCI)} \sqrt{V_{\hat{\theta}_{jl}}}]
\tag{9}
$$

where $t_\alpha^{(FGCI)}$ is the $(1 - \alpha)^{th}$ percentile of the distribution of $T^{(FGCI)}$ developed to apply with the SCIs for $\theta_{jl}$ as follows:

$$
T^{(FGCI)} = \max_{j \neq l} \left| \frac{\hat{\theta}_{jl} - R_{\theta_{jl}}(\boldsymbol{Y}, \boldsymbol{Y}^*, \boldsymbol{\mu}, \boldsymbol{\sigma^2}, \boldsymbol{\delta})}{\sqrt{V_{jl}}} \right|
\tag{10}
$$

Motivated by Hasan and Krishnamoorthy [8], Li et al. [13], Hanning et al. [22], Kharrati-Kopaei and Eftekhar [23], we slightly adjusted the results in [22] to prove the asymptotic coverage probability of the FGCI-based SCI for $\theta_{jl}$. Concerning the properties of the SCI in (9), the simultaneous $100(1 - \alpha)\%$ FGCI for $\theta_{jl}$ follows the asymptotic coverage probability in Theorem 2 in S1 Appendix.

*Algorithm 2: FGCI interval*

```
1 Generate W̄ⱼ*, Sⱼ²* and δ̂ⱼ* being the independent copies of W̄ⱼ, Sⱼ² and δ̂ⱼ,
respectively.
2 Compute (Rμⱼ, Rσⱼ²) and Rδⱼ' in Eqs (6) and (7), respectively.
3 Compute Rθⱼₗ in Eq (8) and T^(FGCI) in Eq (10).
4 Repeat steps 1-3, a large number of times, m = 2500. The empirical
distribution of T^(FGCI) is obtained to compute t₁₋α^(FGCI).
5 Compute the simultaneous 95% FGCI for θⱼₗ, given by Eq (9).
```

## The MOVER interval

The MOVER-based SCI for $\theta_{jl}$ is constructed using the concepts of Zou et al. [28] and Donner and Zou [19]. The construction of $100(1 - \alpha)\%$ MOVER-based SCI for $\theta_{jl} = \theta_j - \theta_l = \exp(\beta_j) - \exp(\beta_l)$ is given by

$$
\begin{aligned}
SCI_{\theta_{jl}}^{(MOVER)} &= [L_{\theta_{ij}}^{(MOVER)}, U_{\theta_{jl}}^{(MOVER)}] \\
&= \left[ \hat{\theta}_{jl} - \sqrt{(\hat{\theta}_j - l_{\theta_j})^2 + (u_{\theta_l} - \hat{\theta}_l)^2}, \hat{\theta}_{jl} + \sqrt{(u_{\theta_j} - \hat{\theta}_j)^2 + (\hat{\theta}_l - l_{\theta_l})^2} \right]
\end{aligned}
$$

where $\hat{\theta}_{jl} = \exp(\hat{\beta}_j) - \exp(\hat{\beta}_l)$. First, the $\theta_j$ is log-transformed as

$$\beta_j = \ln\delta'_j + (\mu_j + \sigma_j^2/2)$$

The estimate of $\beta_j$ is obtained from replacing $\bar{W}_j$, $S_j^2$ and $\hat{\delta}_j$ with their parameters, i.e.
$\hat{\beta}_j = \ln\hat{\delta}'_j + (\bar{W}_j + S_j^2/2); j = 1, 2, .., k$. For considering the constructed CIs for $\theta_j$ and $\theta_l$
denoted as $(l_{\theta_j}, u_{\theta_j})$ and $(l_{\theta_l}, u_{\theta_l})$, the $100(1-\alpha)\%$ MOVER-based CIs for $\beta_j$ and $\beta_l$; $j \neq l$ can be
expressed as

$$
\begin{aligned}
CI_{\beta_j}^{(MOVER)} &= [l_{\beta_j}, u_{\beta_j}] \\
&= \left[ \hat{\beta}_j - \sqrt{(\ln\hat{\delta}'_j - l_{\delta'_j})^2 + (\hat{W}_j + S_j^2/2 - l_{\mu_j,\sigma_j^2})^2}, \right. \\
&\quad \left. \hat{\beta}_j + \sqrt{(u_{\delta'_j} - \ln\hat{\delta}'_j)^2 + (u_{\mu_j,\sigma_j^2} - \hat{W}_j - s_j^2/2)^2} \right]
\end{aligned}
$$

where the $(l_{\delta'_j}, u_{\delta'_j})$ is based on the variance stabilized transformation for $\delta'$, proposed by Das-
Gupta [29] as follows:

$$(l_{\delta'_j}, u_{\delta'_j}) = \ln\left[ \sin^2\left\{ \arcsin\sqrt{\hat{\delta}'_j} \mp \frac{V_{j,1-\alpha/2}}{2\sqrt{n_j}} \right\} \right] \tag{11}$$

where $V_j = 2\sqrt{n_j}(\arcsin\sqrt{\hat{\delta}'_j} - \arcsin\sqrt{\delta'_j}) \sim N(0,1)$; $n_j \to \infty$. Thus, the $100(1-\alpha)\%$ CI
for $\mu_j + \sigma_j^2/2$ becomes

$$
\begin{aligned}
(l_{\mu_j,\sigma_j^2}, u_{\mu_j,\sigma_j^2}) &= \left[ \left(\hat{W}_j + S_j^2/2\right) - \left\{ \left(\frac{T_{j,1-\alpha/2}S_j^2}{n_{j(1)}}\right)^2 + \frac{S_j^4}{4}\left(1 - \frac{n_{j(1)}-1}{\chi_{j,1-\alpha/2,n_{(1)}-1}^2}\right)^2 \right\}^{1/2}, \right. \\
&\quad \left. \left(\hat{W}_j + S_j^2/2\right) + \left\{ \left(\frac{T_{j,1-\alpha/2}S_j^2}{n_{j(1)}}\right)^2 + \frac{S_j^4}{4}\left(\frac{n_{j(1)}-1}{\chi_{j,\alpha/2,n_{(1)}-1}^2} - 1\right)^2 \right\}^{1/2} \right]
\end{aligned} \tag{12}
$$

where $T_j = \sqrt{n_{j(1)}}(\hat{W}_j - \mu_j)/S_j \sim N(0,1)$, and $\chi_{j,n_{j(1)}-1}^2$ is a chi-square distribution with $n_{j(1)}$
$- 1$ degrees of freedom. The $100(1-\alpha)\%$ MOVER interval for $\theta_j$ becomes

$$CI_{\theta_j}^{(MOVER)} = [l_{\theta_j}, u_{\theta_j}] = [\exp\{l_{\beta_j}\}, \exp\{u_{\beta_j}\}]$$

Similarly, we can obtain $CI_{\theta_l}^{(MOVER)} = [l_{\theta_l}, u_{\theta_l}] = [\exp\{l_{\beta_l}\}, \exp\{u_{\beta_l}\}]$. Therefore, the $100(1-\alpha)\%$ MOVER-based SCI for $\theta_{jl}$ is given by

$$
\begin{aligned}
SCI_{\theta_{jl}}^{(MOVER)} &= [L_{\theta_{jl}}^{(MOVER)}, U_{\theta_{jl}}^{(MOVER)}] \\
&= \left[ \hat{\theta}_{jl} - \sqrt{[\hat{\theta}_j - l_{\theta_j}]^2 + [u_{\theta_l} - \hat{\theta}_l]^2}, \right. \\
&\quad \left. \hat{\theta}_{jl} + \sqrt{[u_{\theta_j} - \hat{\theta}_j]^2 + [\hat{\theta}_l - l_{\theta_l}]^2} \right]
\end{aligned} \tag{13}
$$

In accordance with Harvey and Merwe [4], Donner and Zhou [19], Hanning et al. [22],

Thangjai and Niwitpong [30], we slightly adjusted the results of [22] and [30] to prove the MOVER-based SCI in (13) with the asymptotic coverage property, given in Theorem 3 in S1 Appendix.

> *Algorithm 3*: *MOVER interval*
> 1 Compute $(l_{\delta'_j}, u_{\delta'_j})$ and $(l_{\mu_j,\sigma_j^2}, u_{\mu_j,\sigma_j^2})$, as given by Eqs (11) and (12), respectively
> 2 Compute $CI_{\beta_j}^{(MOVER)}$ and $CI_{\beta_l}^{(MOVER)}$.
> 3 Compute 95%SCI-based MOVER for $\theta_{jl}$, as given by Eq (13).

## Bayesian credible intervals

Maneerat et al. [14] proposed highest posterior density (HPD) intervals for the single and difference between two delta-lognormal means based on mixed (normal and inverse chi-square) and uniform priors. For the difference between delta-lognormal means, the HPD-based uniform prior provided the best performance, while that of the HPD-based mixed prior was no different from those of the well-established HPD-based Jeffreys and beta priors. Hence, both are considered and developed here as SCIs for $\theta_{jl}$.

**Mixed prior.** This is a prior of $(\mu_j, \sigma_j^2)$ based on independently drawn samples from normal and inverse chi-square distributions combined with the beta prior of $\delta'_j$ to obtain the mixed prior of $\theta_j$ as follows:

$$
\begin{aligned}
P_M(\mu_j, \sigma_j^2, \delta'_j) \quad &= \frac{\Gamma(a_j)\Gamma(b_j)}{\Gamma(a_j + b_j)} \delta_j'^{a_j-1}(1-\delta'_j)^{b_j-1} \sqrt{\frac{n_{j(1)}^{(0)}}{2\pi}}(\sigma_j^2)^{-1/2} \exp\left\{\frac{k_j^{(0)}}{2\sigma_j^2}(\mu_j^{(0)} - \mu_j)^2\right\} \\
&\quad \frac{(\sigma_j^2)^{-(v_j^{(0)}/2)-1}}{2^{v_j^{(0)}/2}\Gamma(v_j^{(0)}/2)} \exp\left\{\frac{v_j^{(0)}\sigma_j^{2(0)}}{2\sigma_j^2}\right\} \\
&\propto \delta_j'^{a_j-1}(1-\delta'_j)^{b_j-1}(\sigma_j^2)^{-(v_j^{(0)}+1)/2-1} \exp\left\{\frac{1}{2\sigma_j^2}\left[k_j^{(0)}(\mu_j^{(0)} - \mu_j)^2 + v_j^{(0)}\sigma_j^{2(0)}\right]\right\}
\end{aligned}
\tag{14}
$$

where $\delta'_j \sim \text{beta}(a_j, b_j)$, $\mu_j \sim N(\mu_j | \mu_j^{(0)}, \sigma_j^2/k_j^{(0)})$ and $\sigma_j^2 \sim \chi^{-2}(\sigma_j^2 | v_j^{(0)}, \sigma_j^{2(0)})$. The likelihood function is

$$
P(w_j | \mu_j, \sigma_j^2, \delta'_j) = \binom{n_j}{n_{j(1)}} \delta_j'^{n_{j(1)}}(1-\delta'_j)^{n_{j(0)}} \frac{(\sigma^2)^{-n_{j(1)}/2}}{\sqrt{2\pi}} \exp\left\{-\frac{1}{2\sigma_j^2}\sum_{r=1}^{n_{j(1)}}(\ln w_{jr} - \mu_j)^2\right\}
\tag{15}
$$

The posterior of $(\mu_j, \sigma_j^2, \delta'_j)$ derived from the mixed prior (14) and its likelihood (15) becomes

$$
\begin{aligned}
P(\mu_j, \sigma_j^2, \delta'_j | w_j) \propto \quad &\delta_j'^{(n_{j(1)}+a_j)-1}(1-\delta'_j)^{(n_{j(0)}+b_j)-1}(\sigma_j^2)^{-(v_j^{(1)}/2)-1} \\
&\exp\left\{\frac{1}{2\sigma_j^2}\left[k_j^{(1)}(\mu_j^{(1)} - \mu_j)^2 + v_j^{(1)}\sigma_j^{2(1)}\right]\right\}
\end{aligned}
$$

This can be integrated to obtain the respective marginal distributions of $\mu_j, \sigma_j^2$ and $\delta_j'$ as

$$\delta_{j,(M)}'|w_j \quad \sim \text{beta}(n_{j(1)} + a_j, n_{j(0)} + b_j)$$

$$\sigma_{j,(M)}^2|w_j \quad \sim \chi_{v_j^{(1)}}^{-2}(\sigma_j^{2(1)}) \tag{16}$$

$$\mu_{j,(M)}|\sigma_j^2, w_j \quad \sim t_{v_j^{(1)}}(\mu_j|\mu_j^{(1)}, \sigma_j^{2(1)}/k_j^{(1)})$$

where $\mu_j^{(1)} = \bar{w}_j$, $k_j^{(1)} = n_{j(1)}$, $v_j^{(1)} = n_{j(1)} - 1$, and $\sigma_j^{2(1)} = \frac{1}{n_{j(1)}-1}\sum_{r=1}^{n_{j(1)}}(\ln w_{jr} - \bar{w}_j)^2$ are fixed [31]. From Eq (16), the posterior distribution of $\theta_{jl}$ based on the mixed prior is given by

$$\theta_{jl,(M)}^* = \theta_{j,(M)}^* - \theta_{l,(M)}^* \tag{17}$$

where $\theta_{j,(M)}^* = \delta_{j,(M)}' \exp(\mu_{j,(M)} + \sigma_{j,(M)}^2/2)$ and $\theta_{l,(M)}^* = \delta_{l,(M)}' \exp(\mu_{l,(M)} + \sigma_{l,(M)}^2/2)$. According to Ganesh [32], we can define

$$Q \equiv \max_k\{(\hat{\theta}_k - \theta_k^*)|w_k\} - \min_k\{(\hat{\theta}_k - \theta_k^*)|w_k\}$$

where $|(\hat{\theta}_j - \theta_j^*)|w_j - (\hat{\theta}_l - \theta_l^*)|w_l| \leq Q$ for all $j$ and $l$. Thus, we can imply that

$$P(|(\hat{\theta}_j - \theta_j^*) - (\hat{\theta}_l - \theta_l^*)| \leq Q_{\alpha/2}|w) \geq 1 - \alpha$$

Therefore, the simultaneous $100(1 - \alpha)\%$ BCI-M for $\theta_{jl}$ is

$$SBCI_{\theta_{jl}}^{(M)} = \theta_{jl,(M)}^* \pm q_\alpha^{(M)} \tag{18}$$

where $q_\alpha^{(M)}$ denotes $(1 - \alpha)^{th}$ percentile of the distribution of $Q^{(M)} = \max_k\{\theta_{jl,(M)}^*\} - \min_k\{\theta_{jl,(M)}^*\}$.

**Uniform prior.** The uniform prior of $(\mu_j, \log\sigma_j, \delta_j')$ can be written as

$$P_u(\mu_j, \log\sigma_j, \delta_j') \propto constant \tag{19}$$

Meanwhile, the likelihood for $(\mu_j, \log\sigma_j, \delta_j')$ is

$$P(w_j|\mu_j, \log\sigma_j, \delta_j') \quad = P(w_j|\delta_j')P(w_j|\mu_j, \log\sigma_j)$$

$$= \binom{n_j}{n_{j(0)}} \delta_j'^{n_{j(1)}}(1 - \delta_j')^{n_{j(0)}}(\sqrt{2\pi}\log\sigma_j)^{-n_{j(1)}} \tag{20}$$

$$\exp\left\{-\frac{1}{2\sigma_j^2}\left[(n_{j(1)} - 1)s_j^2 + n_{j(1)}(\bar{w}_j - \mu_j)^2\right]\right\}$$

For deriving the first and second derivatives of $P(w_j|\mu_j, \log\sigma_j)$, we obtain

$$I(\mu_j, \log\sigma_j) = \begin{bmatrix} -n_{j(1)}/\sigma_j^2 & 0 \\ 0 & -2n_{j(1)} \end{bmatrix}$$

Using normal approximation, the posterior distribution of $(\mu_j, \log\sigma_j)$ can be approximated as

$$\mu_{j,(U)}|w_j \approx N(\mu_j|\bar{w}_j, s_j^2/n_{j(1)})$$

$$\log\sigma_{j,(U)}|w_j \approx N(\log\sigma_j|\log s_j, [2n_{j(1)}]^{-1})$$

(21)

To transform $\log\sigma$ to $\sigma$, Gelman et al. [33] used a Jacobian transformation to obtain $\sigma_{j,(U)}^2|w_j \approx N(\sigma_j^2|\tilde{\sigma}_j^2, 2\tilde{\sigma}_j^4(n_{j(1)}+2)^{-1})$, with $\tilde{\sigma}_j^2 = n_{j(1)}s_j^2/(n_{j(1)}+2)$. Since $(\mu_j, \log\sigma_j)$ and $\delta_j'$ are independent. To focus on the posterior distribution of $\delta_j'$, the uniform prior in Eq (19) can be combined with its likelihood (20) to yield

$$P(\delta_j') \propto \delta_j'^{(n_{j(1)}+a_j)-1}(1-\delta')_j^{(n_{j(0)}+b_j)-1}$$

(22)

which $\delta_{j,(U)}'|w_j \sim \text{beta}(n_{j(1)}+a_j, n_{j(0)}+b_j)$; $a_j = b_j = 1$. From Eqs (21) and (22), the posterior of $\theta_{jl}$ based on the uniform prior can be written as

$$\theta_{jl,(U)}^* = \theta_{j,(U)}^* - \theta_{l,(U)}^*$$

(23)

where $\theta_{j,(U)}^* = \delta_{j,(U)}' \exp(\mu_{j,(U)} + \sigma_{j,(U)}^2/2)$ and $\theta_{l,(U)}^* = \delta_{l,(U)}' \exp(\mu_{l,(U)} + \sigma_{l,(U)}^2/2)$. Hence, the simultaneous $100(1-\alpha)\%$ BCI-U for $\theta_{jl}$ is given by

$$SBCI_{\theta_{jl}}^{(U)} = \theta_{jl,(U)}^* \pm q_\alpha^{(U)}$$

(24)

where $q_\alpha^{(U)}$ denotes the $(1-\alpha)^{th}$ percentile of the distribution of $Q^{(U)} = \max_k\{\theta_{jl,(U)}^*\} - \min_k\{\theta_{jl,(U)}^*\}$.

*Algorithm 4*: SBCIs-based mixed and uniform priors

```
1 Mixed prior
   1.1 Generate δ'ⱼ,₍ₘ₎|wⱼ, σ²ⱼ,₍ₘ₎|wⱼ and μⱼ,₍ₘ₎|σ²ⱼ,wⱼ, given in Eq (16) being the
marginal posterior distributions of δ'ⱼ, σ²ⱼ and μⱼ, respectively.
   1.2 Compute θ*ⱼₗ,₍ₘ₎ in Eq (17).
   1.3 Repeat steps 1.1-1.2, a large number of times, m = 2500. The
empirical distribution of Q₍ₘ₎α is obtained to compute q₍ₘ₎α.
2 Uniform prior
   2.1 Generate δ'ⱼ,₍ᵤ₎|wⱼ, σ²ⱼ,₍ᵤ₎|wⱼ and μⱼ,₍ᵤ₎|σ²ⱼ,wⱼ being the marginal posterior
distributions of δ'ⱼ, σ²ⱼ and μⱼ, respectively.
   2.2 Compute θ*ⱼₗ,₍ᵤ₎ in Eq (23).
   2.3 Repeat steps 2.1-2.2, a large number of times, m = 2500. The
empirical distribution of Q₍ᵤ₎α is obtained to compute q₍ᵤ₎α.
3 Compute 95%SBCIs-based mixed and uniform priors for θⱼₗ given in Eqs
(18) and (24), respectively.
```

## Simulation studies

Monte Carlo simulations were conducted to examine the performances of the SCIs by considering their coverage probabilities (CPs), lower and upper error probabilities (LEP and UEP, respectively), and relative average lengths (RALs; the ratio between the average lengths of each proposed SCI and MOVER). Note that the average lengths of MOVER used in the comparison were reported. According to the performance measures for SCIs used by Li et al. [13], the best-performing method is where the CP is close to or greater than the nominal level $(1-\alpha = 95\%)$ and the RAL is less than 1 and the smallest, while the required values of LEP and UEP are balanced at 2.5%. The parameter combinations in the simulation studies were set as follows:

- Sample cases: $k = 3, 5$

- Sample size: $\boldsymbol{n} = (n_1, n_2, \ldots, n_k)$

- Population variance: $\boldsymbol{\sigma^2} = (\sigma_1^2, \sigma_2^2, \ldots, \sigma_k^2)$

- Population mean: $\boldsymbol{\mu} = \mu_1, \ldots, = \mu_k = 0$

- Probability of having zero: $\boldsymbol{\delta} = (\delta_1, \delta_2, \ldots, \delta_k)$

These parameters were fixed and varied in different scenarios, as reported in Tables 1 and 2. Random samples were drawn from a delta-lognormal distribution for each parameter combination. The simulation method for PB differed from the other methods in that repeated samples were drawn from the original random samples, as mentioned above. Algorithm 5 shows the simulation procedure with the steps to compute the SCI performances.

*Algorithm 5*:

```
1 Generate a random sample W_j ~ Δ(μ_j,σ²_j,δ_j); j = 1, 2, ..., k. Compute w̄_j,
```
$s_j^2$, and $\hat{\delta}_j' = 1 - \hat{\delta}_j$ are the observed values of $\bar{W}_j$, $S_j^2$ and $\hat{\delta}_j'$
```
2 Compute the SCIs-based the following methods:
  2.1 PB, FGCI and MOVER from Algorithms 1,2,3 and record whether or
not all the values of θ_jl are in their corresponding
```
$SCI_{\theta_{jl}}^{(PB)}$, $SCI_{\theta_{jl}}^{(MOVER)}$ and $SCI_{\theta_{jl}}^{(FGCI)}$, respectively.
```
  2.2 BCI-M and BCI-U from Algorithm 4, and record whether or not all
the values of θ_jl are in their corresponding
```
$SCI_{\theta_{jl}}^{(BCI-M)}$ and $SCI_{\theta_{jl}}^{(BCI-U)}$, respectively.
```
3 Repeat step 2 with a number of times M = 5000, so the time proportion
that all θ_jl are in their corresponding SCIs giving the estimated CP.
```

## Results

### Simulation results

R, version 4.0.4 was used for the computations of the simulations and the application. For sample case $k = 3$ (Table 1 and Fig 1), the numerical evaluations show that BCI-U provided the correct CP with the shortest interval for small-to-large differences in $\sigma_j^2$ and $\delta_i$ and equal sample sizes. For unequal sample sizes, PB obtained good performance criteria results when the difference in $\sigma_i^2$ was large, while BCI-U and MOVER performed better for the rest of the cases. Likewise, the balance between LEP and UEP was maintained by MOVER. Both BCI-M and FGCI attained good CPs but wider average lengths than the others. For $k = 5$ (Table 2 and Fig 2), the simulation results reveal that MOVER provided good and stable performances for small differences in $\sigma^2$ and $\delta_i$, while importantly, also provided a good balance between LEP and UEP. The performance of PB was satisfactory for all large differences in $\sigma^2$ whereas BCI-M, BCI-U, and FGCI performed poorly, with higher CPs and wider ALs than the others).

### An example using real data

This provides an illustrative example of applying the proposed SCIs to analyze natural rainfall datasets including extreme weather events in Thailand. There are five areas (northern, northeastern, central, eastern, and southern (east and west coasts)) and three seasons (summer, rainy, and winter) in Thailand, for which we considered natural rainfall datasets for July 2018. Table 3 provides data on the weekly natural rainfall amounts recorded by the Thai Meteorological Department: north (62 substations), northeast (210 substations), central (57 substations), east (29 substations), southeast (89 substations), and southwest (30 substations). There

**Table 1. Performance metrics for the 95% SCIs for $\theta_{ji}$; 3 sample cases.**

| Sample case $k = 3$ | | | PB | | | BCI-M | | | BCI-U | | | FGCI | | | MOVER | | | RAL | | | | |
| n | $\sigma^2$ | $\delta$ | LEP | CP | UEP | LEP | CP | UEP | LEP | CP | UEP | LEP | CP | UEP | LEP | CP | UEP | PB | BCI-M | BCI-U | FGCI | MOVER |
|---|---|---|---|---|---|---|---|---|---|---|---|---|---|---|---|---|---|---|---|---|---|---|
| (20,20,20) | (1,2,3) | (0.1,0.2,0.3) | 3.39 | 96.61 | 0.00 | 0.12 | 99.88 | 0.00 | 3.59 | 96.41 | 0.00 | 0.06 | 99.94 | 0.00 | 2.02 | 95.79 | 2.19 | 0.214 | 1.351 | **0.201** | 1.204 | * |
| | | (0.1,0.3,0.5) | 2.65 | 97.35 | 0.00 | 0.09 | 99.91 | 0.00 | 2.45 | 97.55 | 0.00 | 0.06 | 99.94 | 0.00 | 1.94 | 95.43 | 2.63 | 0.004 | 0.308 | **0.004** | 0.221 | * |
| | | (0.3,0.5,0.5) | 3.06 | 96.94 | 0.00 | 0.07 | 99.93 | 0.00 | 3.33 | 96.67 | 0.00 | 0.01 | 99.99 | 0.00 | 1.80 | 96.26 | 1.94 | 2.7e-4 | 0.080 | **2.8e-4** | 0.052 | * |
| | (3,5,7) | (0.1,0.2,0.3) | 6.92 | 93.08 | 0.00 | 0.07 | 99.93 | 0.00 | 3.87 | 96.13 | 0.00 | 0.06 | 99.94 | 0.00 | 2.42 | 96.03 | 1.55 | 9.7e-5 | 0.182 | **2.5e-4** | 0.110 | * |
| | | (0.1,0.3,0.5) | 6.36 | 93.64 | 0.00 | 0.04 | 99.96 | 0.00 | 3.54 | 96.46 | 0.00 | 0.03 | 99.97 | 0.00 | 2.16 | 96.18 | 1.66 | 1.6e-11 | 5e-4 | **8.1e-11** | 4e-4 | * |
| | | (0.3,0.5,0.5) | 6.15 | 93.85 | 0.00 | 0.03 | 99.97 | 0.00 | 4.15 | 95.85 | 0.00 | 0.05 | 99.95 | 0.00 | 2.14 | 96.33 | 1.53 | 8.4e-12 | 0.001 | **6.0e-11** | 0.001 | * |
| (50,50,50) | (1,2,3) | (0.1,0.2,0.3) | 3.13 | 96.87 | 0.00 | 1.00 | 99.00 | 0.00 | 4.49 | 95.51 | 0.00 | 0.60 | 99.40 | 0.00 | 1.96 | 95.71 | 2.33 | 1.031 | 1.709 | **0.952** | 1.762 | * |
| | | (0.1,0.3,0.5) | 2.91 | 97.09 | 0.00 | 0.59 | 99.41 | 0.00 | 3.21 | 96.79 | 0.00 | 0.43 | 99.57 | 0.00 | 1.95 | 95.52 | 2.53 | 0.828 | 1.773 | **0.805** | 1.740 | * |
| | | (0.3,0.5,0.5) | 3.13 | 96.87 | 0.00 | 0.70 | 99.30 | 0.00 | 3.84 | 96.16 | 0.00 | 0.33 | 99.67 | 0.00 | 1.78 | 95.73 | 2.49 | 0.810 | 1.786 | **0.788** | 1.805 | * |
| | (3,5,7) | (0.1,0.2,0.3) | 6.76 | 93.24 | 0.00 | 0.49 | 99.51 | 0.00 | 4.17 | 95.83 | 0.00 | 0.54 | 99.46 | 0.00 | 2.49 | 95.04 | 2.47 | 0.338 | 1.793 | **0.544** | 1.571 | * |
| | | (0.1,0.3,0.5) | 6.49 | 93.51 | 0.00 | 0.32 | 99.68 | 0.00 | 3.15 | 96.85 | 0.00 | 0.42 | 99.58 | 0.00 | 2.05 | 95.57 | 2.38 | 0.096 | 1.257 | **0.178** | 0.979 | * |
| | | (0.3,0.5,0.5) | 6.41 | 93.59 | 0.00 | 0.31 | 99.69 | 0.00 | 3.71 | 96.29 | 0.00 | 0.29 | 99.71 | 0.00 | 2.19 | 95.36 | 2.45 | 0.088 | 1.226 | **0.169** | 1.056 | * |
| (100,100,100) | (1,2,3) | (0.1,0.2,0.3) | 2.47 | 97.53 | 0.00 | 2.34 | 97.66 | 0.00 | 4.81 | 95.17 | 0.02 | 0.96 | 99.04 | 0.00 | 1.96 | 95.32 | 2.72 | 1.242 | 1.522 | 1.110 | 1.632 | ** |
| | | (0.1,0.3,0.5) | 2.25 | 97.75 | 0.00 | 1.37 | 98.63 | 0.00 | 3.71 | 96.27 | 0.03 | 0.73 | 99.27 | 0.00 | 1.75 | 95.44 | 2.81 | 1.152 | 1.590 | 1.061 | 1.662 | ** |
| | | (0.3,0.5,0.5) | 2.57 | 97.43 | 0.00 | 1.55 | 98.45 | 0.00 | 4.49 | 95.50 | 0.01 | 0.71 | 99.29 | 0.00 | 1.79 | 95.41 | 2.80 | 1.147 | 1.617 | 1.071 | 1.702 | ** |
| | (3,5,7) | (0.1,0.2,0.3) | 5.46 | 94.54 | 0.00 | 1.06 | 98.94 | 0.00 | 4.25 | 95.75 | 0.00 | 0.87 | 99.13 | 0.00 | 2.24 | 95.31 | 2.45 | 0.862 | 2.068 | **1.229** | 1.907 | * |
| | | (0.1,0.3,0.5) | 6.21 | 93.79 | 0.00 | 0.86 | 99.14 | 0.00 | 3.71 | 96.29 | 0.00 | 0.83 | 99.17 | 0.00 | 2.25 | 95.30 | 2.45 | 0.629 | 2.048 | **0.975** | 1.767 | * |
| | | (0.3,0.5,0.5) | 6.15 | 93.85 | 0.00 | 0.84 | 99.16 | 0.00 | 4.19 | 95.81 | 0.00 | 0.70 | 99.30 | 0.00 | 2.05 | 95.18 | 2.77 | 0.611 | 2.040 | **0.954** | 1.860 | * |
| (20,50,100) | (1,2,3) | (0.1,0.2,0.3) | 1.40 | 98.60 | 0.00 | 0.27 | 99.73 | 0.00 | 1.91 | 98.08 | 0.01 | 0.09 | 99.91 | 0.00 | 2.07 | 95.84 | 2.09 | 0.887 | 1.202 | **0.824** | 1.496 | ** |
| | | (0.1,0.3,0.5) | 0.88 | 99.12 | 0.00 | 0.06 | 99.94 | 0.00 | 1.31 | 98.69 | 0.00 | 0.01 | 99.99 | 0.00 | 1.58 | 96.18 | 2.24 | 0.824 | 1.252 | **0.793** | 1.526 | ** |
| | | (0.3,0.5,0.5) | 1.07 | 98.93 | 0.00 | 0.09 | 99.91 | 0.00 | 1.74 | 98.26 | 0.00 | 0.02 | 99.98 | 0.00 | 1.73 | 96.58 | 1.69 | 0.767 | 1.262 | **0.733** | 1.557 | ** |
| | (3,5,7) | (0.1,0.2,0.3) | 3.77 | 96.23 | 0.00 | 0.15 | 99.85 | 0.00 | 2.01 | 97.99 | 0.00 | 0.07 | 99.93 | 0.00 | 2.07 | 95.75 | 2.17 | **0.500** | 1.501 | 0.726 | 1.785 | * |
| | | (0.1,0.3,0.5) | 3.20 | 96.80 | 0.00 | 0.06 | 99.94 | 0.00 | 1.33 | 98.67 | 0.00 | 0.06 | 99.94 | 0.00 | 2.05 | 95.34 | 2.38 | **0.413** | 1.590 | 0.637 | 1.848 | * |
| | | (0.3,0.5,0.5) | 3.34 | 96.66 | 0.00 | 0.05 | 99.95 | 0.00 | 1.78 | 98.22 | 0.00 | 0.03 | 99.97 | 0.00 | 2.21 | 96.15 | 1.92 | **0.217** | 1.165 | 0.329 | 1.387 | * |
| (50,100,200) | (1,2,3) | (0.1,0.2,0.3) | 2.11 | 97.89 | 0.00 | 1.27 | 98.73 | 0.00 | 2.99 | 96.79 | 0.22 | 0.35 | 99.65 | 0.00 | 1.80 | 95.94 | 2.26 | 1.101 | 1.163 | **0.977** | 1.439 | ** |
| | | (0.1,0.3,0.5) | 1.65 | 98.35 | 0.00 | 0.60 | 99.40 | 0.00 | 2.17 | 97.75 | 0.09 | 0.15 | 99.85 | 0.00 | 1.73 | 95.56 | 2.71 | 1.047 | 1.181 | **0.948** | 1.449 | ** |
| | | (0.3,0.5,0.5) | 1.38 | 98.62 | 0.00 | 0.71 | 99.29 | 0.00 | 2.50 | 97.47 | 0.03 | 0.15 | 99.85 | 0.00 | 1.60 | 96.28 | 2.12 | 1.032 | 1.202 | **0.941** | 1.475 | ** |
| | (3,5,7) | (0.1,0.2,0.3) | 5.20 | 94.80 | 0.00 | 0.89 | 99.11 | 0.00 | 3.27 | 96.73 | 0.00 | 0.47 | 99.53 | 0.00 | 2.08 | 95.75 | 2.17 | **0.909** | 1.687 | 1.249 | 1.839 | * |
| | | (0.1,0.3,0.5) | 4.86 | 95.14 | 0.00 | 0.35 | 99.65 | 0.00 | 2.30 | 97.70 | 0.00 | 0.26 | 99.74 | 0.00 | 2.28 | 95.34 | 2.38 | **0.816** | 1.740 | 1.176 | 1.847 | * |
| | | (0.3,0.5,0.5) | 4.62 | 95.38 | 0.00 | 0.35 | 99.65 | 0.00 | 2.33 | 97.67 | 0.00 | 0.17 | 99.83 | 0.00 | 1.93 | 96.15 | 1.92 | **0.776** | 1.715 | 1.111 | 1.921 | * |
| (100,100,200) | (1,2,3) | (0.1,0.2,0.3) | 2.11 | 97.89 | 0.00 | 1.43 | 98.57 | 0.00 | 3.25 | 96.60 | 0.15 | 0.78 | 99.22 | 0.00 | 2.05 | 95.17 | 2.78 | 1.229 | 1.248 | 1.040 | 1.519 | ** |
| | | (0.1,0.3,0.5) | 1.95 | 98.05 | 0.00 | 0.93 | 99.07 | 0.00 | 2.34 | 97.61 | 0.05 | 0.62 | 99.38 | 0.00 | 1.75 | 95.61 | 2.64 | 1.177 | 1.275 | 1.011 | 1.543 | ** |
| | | (0.3,0.5,0.5) | 1.86 | 98.14 | 0.00 | 0.81 | 99.19 | 0.00 | 2.47 | 97.51 | 0.02 | 0.41 | 99.59 | 0.00 | 1.65 | 95.98 | 2.37 | 1.165 | 1.306 | 1.014 | 1.578 | ** |
| | (3,5,7) | (0.1,0.2,0.3) | 5.22 | 94.78 | 0.00 | 1.13 | 98.87 | 0.00 | 3.01 | 96.99 | 0.00 | 0.79 | 99.21 | 0.00 | 2.10 | 95.25 | 2.65 | **1.041** | 1.816 | 1.350 | 1.949 | * |
| | | (0.1,0.3,0.5) | 5.30 | 94.70 | 0.00 | 0.68 | 99.32 | 0.00 | 2.31 | 97.69 | 0.00 | 0.67 | 99.33 | 0.00 | 2.26 | 95.18 | 2.56 | **0.934** | 1.855 | 1.255 | 1.947 | * |
| | | (0.3,0.5,0.5) | 5.01 | 94.99 | 0.00 | 0.69 | 99.31 | 0.00 | 2.59 | 97.41 | 0.00 | 0.41 | 99.59 | 0.00 | 1.85 | 95.84 | 2.31 | **0.889** | 1.866 | 1.213 | 2.040 | * |

**Note:** Bold denotes the best-performing method.

*MOVER satisfies the CP criteria, and

**it is also the best-performing method.

**Table 2. Performance metrics for the 95% SCIs for $\theta_{ji}$; 5 sample cases.**

Sample case $k = 5$

| n | $\sigma^2$ | $\delta$ | PB LEP | PB CP | PB UEP | BCI-M LEP | BCI-M CP | BCI-M UEP | BCI-U LEP | BCI-U CP | BCI-U UEP | FGCI LEP | FGCI CP | FGCI UEP | MOVER LEP | MOVER CP | MOVER UEP | PB (RAL) | BCI-M (RAL) | BCI-U (RAL) | FGCI (RAL) | MOVER (RAL) |
|---|---|---|---|---|---|---|---|---|---|---|---|---|---|---|---|---|---|---|---|---|---|---|
| (20,50,100,200) | (0.5,1,1.5,2,3) | (0.1,0.2,0.2,0.3,0.3) | 0.75 | 99.25 | 0.00 | 0.19 | 99.81 | 0.00 | 0.50 | 99.49 | 0.01 | 0.10 | 99.90 | 0.00 | 2.14 | 95.52 | 2.34 | 1.170 | 1.628 | 1.373 | 1.586 | ** |
| | | (0.1,0.2,0.3,0.5,0.7) | 0.36 | 99.63 | 0.01 | 0.01 | 99.99 | 0.00 | 0.10 | 99.90 | 0.00 | 0.02 | 99.98 | 0.00 | 1.67 | 96.04 | 2.28 | 1.052 | 1.669 | 1.246 | 1.616 | ** |
| | | (0.3,0.5,0.5,0.7,0.7) | 0.30 | 99.70 | 0.00 | 0.01 | 99.99 | 0.00 | 0.12 | 99.88 | 0.00 | 0.01 | 99.99 | 0.00 | 1.54 | 96.57 | 1.89 | 0.993 | 1.733 | 1.197 | 1.698 | ** |
| | (1,2,2,5,5) | (0.1,0.2,0.2,0.3,0.3) | 1.87 | 98.13 | 0.00 | 0.06 | 99.94 | 0.00 | 0.31 | 99.69 | 0.00 | 0.31 | 99.69 | 0.00 | 2.11 | 95.63 | 2.26 | 1.064 | 2.242 | 1.543 | 1.801 | ** |
| | | (0.1,0.2,0.3,0.5,0.7) | 2.09 | 97.90 | 0.01 | 0.05 | 99.95 | 0.00 | 0.55 | 99.44 | 0.01 | 0.17 | 99.83 | 0.00 | 2.06 | 95.60 | 2.35 | **0.876** | 2.357 | 1.346 | 1.808 | ** |
| | | (0.3,0.5,0.5,0.7,0.7) | 2.19 | 97.81 | 0.00 | 0.03 | 99.97 | 0.00 | 0.27 | 99.73 | 0.00 | 0.08 | 99.92 | 0.00 | 2.16 | 95.69 | 2.15 | **0.581** | 2.330 | 0.940 | 1.649 | * |
| | (1,3,5,7,10) | (0.1,0.2,0.2,0.3,0.3) | 3.38 | 96.62 | 0.00 | 0.63 | 99.37 | 0.00 | 1.82 | 98.18 | 0.00 | 0.38 | 99.62 | 0.00 | 2.22 | 95.48 | 2.30 | **0.956** | 2.829 | 1.981 | 2.143 | * |
| | | (0.1,0.2,0.3,0.5,0.7) | 3.35 | 96.65 | 0.00 | 0.17 | 99.83 | 0.00 | 1.16 | 98.84 | 0.00 | 0.33 | 99.67 | 0.00 | 2.15 | 95.48 | 2.37 | **0.451** | 2.528 | 1.097 | 1.584 | * |
| | | (0.3,0.5,0.5,0.7,0.7) | 3.26 | 96.74 | 0.00 | 0.14 | 99.86 | 0.00 | 1.36 | 98.64 | 0.00 | 0.16 | 99.84 | 0.00 | 1.90 | 95.99 | 2.11 | **0.349** | 2.303 | 0.833 | 1.789 | * |
| | (3,3,5,7,10) | (0.1,0.2,0.2,0.3,0.3) | 3.79 | 96.21 | 0.00 | 0.47 | 99.53 | 0.00 | 1.81 | 98.19 | 0.00 | 0.12 | 99.88 | 0.00 | 2.30 | 95.69 | 2.01 | **0.798** | 2.723 | 1.851 | 2.572 | * |
| | | (0.1,0.2,0.3,0.5,0.7) | 3.63 | 96.37 | 0.00 | 0.08 | 99.92 | 0.00 | 1.07 | 98.93 | 0.00 | 0.14 | 99.86 | 0.00 | 2.05 | 96.14 | 1.82 | **0.387** | 2.517 | 1.065 | 1.884 | * |
| | | (0.3,0.5,0.5,0.7,0.7) | 3.13 | 96.87 | 0.00 | 0.09 | 99.91 | 0.00 | 1.41 | 98.59 | 0.00 | 0.05 | 99.95 | 0.00 | 1.85 | 96.58 | 1.58 | **0.312** | 2.335 | 0.796 | 2.260 | * |
| (50,100,200,200,200) | (0.5,1,1.5,2,3) | (0.1,0.2,0.2,0.3,0.3) | 0.79 | 99.19 | 0.02 | 0.60 | 99.40 | 0.00 | 1.06 | 98.93 | 0.01 | 0.28 | 99.72 | 0.00 | 1.92 | 95.66 | 2.42 | 1.257 | 1.812 | 1.570 | 1.495 | ** |
| | | (0.1,0.2,0.3,0.5,0.7) | 0.74 | 99.26 | 0.00 | 0.19 | 99.81 | 0.00 | 0.47 | 99.53 | 0.00 | 0.17 | 99.83 | 0.00 | 1.78 | 95.61 | 2.61 | 1.145 | 1.862 | 1.413 | 1.531 | ** |
| | | (0.3,0.5,0.5,0.7,0.7) | 0.74 | 99.26 | 0.00 | 0.17 | 99.83 | 0.00 | 0.61 | 99.39 | 0.00 | 0.10 | 99.90 | 0.00 | 1.61 | 96.30 | 2.09 | 1.115 | 1.914 | 1.420 | 1.564 | ** |
| | (1,2,2,5,5) | (0.1,0.2,0.2,0.5,0.6) | 1.78 | 98.22 | 0.01 | 0.05 | 99.95 | 0.00 | 0.20 | 99.80 | 0.00 | 0.60 | 99.40 | 0.00 | 2.16 | 95.44 | 2.40 | 1.204 | 2.132 | 1.703 | 1.662 | ** |
| | | (0.1,0.2,0.5,0.5,0.6) | 1.76 | 98.24 | 0.01 | 0.03 | 99.97 | 0.00 | 0.27 | 99.72 | 0.00 | 0.31 | 99.69 | 0.00 | 1.85 | 95.52 | 2.63 | 1.043 | 2.274 | 1.531 | 1.708 | ** |
| | | (0.1,0.5,0.5,0.5,0.6) | 2.19 | 97.80 | 0.00 | 0.03 | 99.97 | 0.00 | 0.24 | 99.76 | 0.00 | 0.28 | 99.72 | 0.00 | 1.91 | 95.86 | 2.23 | **0.960** | 2.342 | 1.437 | 1.750 | * |
| | (1,3,5,7,10) | (0.1,0.2,0.2,0.5,0.6) | 3.64 | 96.36 | 0.00 | 1.10 | 98.90 | 0.00 | 2.52 | 97.48 | 0.00 | 0.69 | 99.31 | 0.00 | 2.13 | 95.41 | 2.46 | **0.926** | 3.098 | 2.247 | 1.771 | * |
| | | (0.1,0.2,0.5,0.5,0.6) | 3.57 | 96.43 | 0.00 | 0.54 | 99.46 | 0.00 | 1.95 | 98.05 | 0.00 | 0.80 | 99.20 | 0.00 | 2.31 | 95.08 | 2.62 | **0.403** | 2.603 | 1.141 | 1.261 | * |
| | | (0.1,0.5,0.5,0.5,0.6) | 3.71 | 96.29 | 0.00 | 0.60 | 99.40 | 0.00 | 2.33 | 97.67 | 0.00 | 0.53 | 99.47 | 0.00 | 2.18 | 95.34 | 2.47 | **0.423** | 2.593 | 1.139 | 1.423 | * |
| | (3,3,5,7,10) | (0.1,0.2,0.2,0.3,0.3) | 4.73 | 95.27 | 0.00 | 1.29 | 98.71 | 0.00 | 2.62 | 97.38 | 0.00 | 0.58 | 99.42 | 0.00 | 2.25 | 95.24 | 2.51 | **0.816** | 3.070 | 2.231 | 1.821 | * |
| | | (0.1,0.2,0.3,0.5,0.7) | 4.44 | 95.56 | 0.00 | 0.46 | 99.54 | 0.00 | 1.92 | 98.08 | 0.00 | 0.60 | 99.40 | 0.00 | 2.07 | 95.65 | 2.28 | **0.341** | 2.578 | 1.112 | 1.265 | * |
| | | (0.3,0.5,0.5,0.7,0.7) | 4.18 | 95.82 | 0.00 | 0.47 | 99.53 | 0.00 | 2.33 | 97.67 | 0.00 | 0.45 | 99.55 | 0.00 | 2.08 | 95.55 | 2.36 | **0.365** | 2.595 | 1.135 | 1.447 | * |
| (100,100,100,200,200) | (0.5,1,1.5,2,3) | (0.1,0.2,0.2,0.5,0.6) | 0.72 | 99.28 | 0.00 | 0.48 | 99.52 | 0.00 | 0.85 | 99.14 | 0.00 | 0.28 | 99.72 | 0.00 | 1.89 | 95.62 | 2.49 | 1.301 | 1.772 | 1.534 | 1.538 | ** |
| | | (0.1,0.2,0.5,0.5,0.6) | 0.68 | 99.32 | 0.00 | 0.07 | 99.93 | 0.00 | 0.27 | 99.73 | 0.00 | 0.15 | 99.85 | 0.00 | 1.70 | 95.75 | 2.55 | 1.200 | 1.822 | 1.390 | 1.581 | ** |
| | | (0.1,0.5,0.5,0.5,0.6) | 0.70 | 99.30 | 0.00 | 0.09 | 99.91 | 0.00 | 0.39 | 99.61 | 0.00 | 0.11 | 99.89 | 0.00 | 1.62 | 96.09 | 2.29 | 1.161 | 1.873 | 1.392 | 1.623 | ** |
| | (1,2,2,5,5) | (0.1,0.2,0.2,0.5,0.6) | 1.62 | 98.38 | 0.00 | 0.10 | 99.90 | 0.00 | 0.27 | 99.73 | 0.00 | 0.46 | 99.54 | 0.00 | 2.03 | 95.43 | 2.54 | 1.245 | 2.111 | 1.690 | 1.699 | ** |
| | | (0.1,0.2,0.5,0.5,0.6) | 1.71 | 98.29 | 0.00 | 0.06 | 99.94 | 0.00 | 0.26 | 99.74 | 0.00 | 0.27 | 99.73 | 0.00 | 1.88 | 95.53 | 2.59 | 1.086 | 2.243 | 1.519 | 1.745 | ** |
| | | (0.1,0.5,0.5,0.5,0.6) | 1.89 | 98.11 | 0.00 | 0.02 | 99.98 | 0.00 | 0.27 | 99.73 | 0.00 | 0.24 | 99.76 | 0.00 | 1.93 | 95.68 | 2.40 | **0.991** | 2.325 | 1.427 | 1.786 | * |
| | (1,3,5,7,10) | (0.1,0.2,0.2,0.5,0.6) | 3.50 | 96.50 | 0.00 | 1.22 | 98.78 | 0.00 | 2.65 | 97.35 | 0.00 | 0.63 | 99.37 | 0.00 | 2.34 | 95.14 | 2.52 | **0.989** | 3.040 | 2.210 | 1.884 | * |
| | | (0.1,0.2,0.5,0.5,0.6) | 3.25 | 96.75 | 0.00 | 0.30 | 99.70 | 0.00 | 1.76 | 98.24 | 0.00 | 0.51 | 99.49 | 0.00 | 2.19 | 95.18 | 2.64 | **0.451** | 2.602 | 1.151 | 1.333 | * |
| | | (0.1,0.5,0.5,0.5,0.6) | 3.30 | 96.70 | 0.00 | 0.30 | 99.70 | 0.00 | 2.00 | 98.00 | 0.00 | 0.31 | 99.69 | 0.00 | 1.91 | 95.61 | 2.48 | **0.472** | 2.598 | 1.151 | 1.531 | * |
| | (3,3,5,7,10) | (0.1,0.2,0.2,0.3,0.3) | 4.87 | 95.13 | 0.00 | 1.19 | 98.81 | 0.00 | 2.78 | 97.22 | 0.00 | 0.61 | 99.39 | 0.00 | 2.20 | 95.50 | 2.30 | **0.855** | 3.033 | 2.204 | 1.856 | * |
| | | (0.1,0.2,0.3,0.5,0.7) | 4.45 | 95.55 | 0.00 | 0.38 | 99.62 | 0.00 | 1.95 | 98.05 | 0.00 | 0.56 | 99.44 | 0.00 | 2.20 | 95.25 | 2.55 | **0.358** | 2.570 | 1.118 | 1.287 | * |
| | | (0.3,0.5,0.5,0.7,0.7) | 4.08 | 95.92 | 0.00 | 0.38 | 99.62 | 0.00 | 1.96 | 98.04 | 0.00 | 0.33 | 99.67 | 0.00 | 1.86 | 95.72 | 2.42 | **0.377** | 2.576 | 1.123 | 1.479 | * |

*(Continued)*

**Table 2.** (Continued)

| | | | PB | | | BCI-M | | | BCI-U | | | FGCI | | | MOVER | | | RAL | | | | |
| | | | LEP | CP | UEP | LEP | CP | UEP | LEP | CP | UEP | LEP | CP | UEP | LEP | CP | UEP | PB | BCI-M | BCI-U | FGCI | MOVER |
|---|---|---|---|---|---|---|---|---|---|---|---|---|---|---|---|---|---|---|---|---|---|---|
| n | $\sigma^2$ | $\delta$ | | | | | | | | | | | | | | | | | | | | |
| (100,100,200,200,300) | (0.5,1,1.5,2,3) | (0.1,0.2,0.2,0.5,0.6) | 0.70 | 99.30 | 0.00 | 0.47 | 99.53 | 0.00 | 0.76 | 99.24 | 0.00 | 0.25 | 99.75 | 0.00 | 1.91 | 95.53 | 2.56 | 1.300 | 1.685 | 1.530 | 1.495 | ** |
| | | (0.1,0.2,0.5,0.5,0.6) | 0.53 | 99.47 | 0.00 | 0.08 | 99.92 | 0.00 | 0.23 | 99.76 | 0.01 | 0.12 | 99.88 | 0.00 | 1.58 | 96.00 | 2.43 | 1.233 | 1.685 | 1.400 | 1.534 | ** |
| | | (0.1,0.5,0.5,0.5,0.6) | 0.61 | 99.39 | 0.00 | 0.13 | 99.87 | 0.00 | 0.41 | 99.58 | 0.00 | 0.13 | 99.87 | 0.00 | 1.66 | 96.16 | 2.18 | 1.194 | 1.739 | 1.408 | 1.573 | ** |
| | (1,2,2,5,5) | (0.1,0.2,0.2,0.5,0.6) | 1.56 | 98.43 | 0.01 | 0.07 | 99.93 | 0.00 | 0.26 | 99.74 | 0.00 | 0.54 | 99.46 | 0.00 | 2.14 | 95.23 | 2.63 | 1.245 | 2.104 | 1.728 | 1.643 | ** |
| | | (0.1,0.2,0.5,0.5,0.6) | 1.78 | 98.21 | 0.01 | 0.13 | 99.87 | 0.00 | 0.35 | 99.64 | 0.01 | 0.39 | 99.61 | 0.00 | 2.04 | 95.34 | 2.62 | 1.152 | 2.234 | 1.638 | 1.701 | ** |
| | | (0.1,0.5,0.5,0.5,0.6) | 1.86 | 98.14 | 0.00 | 0.06 | 99.94 | 0.00 | 0.30 | 99.70 | 0.00 | 0.31 | 99.69 | 0.00 | 1.86 | 95.58 | 2.56 | 1.035 | 2.309 | 1.507 | 1.723 | ** |
| | (1,3,5,7,10) | (0.1,0.2,0.2,0.5,0.6) | 3.45 | 96.55 | 0.00 | 1.06 | 98.94 | 0.00 | 2.17 | 97.83 | 0.00 | 0.60 | 99.40 | 0.00 | 2.20 | 95.33 | 2.48 | 1.105 | 3.064 | 2.467 | 1.918 | ** |
| | | (0.1,0.2,0.5,0.5,0.6) | 2.98 | 97.02 | 0.00 | 0.42 | 99.58 | 0.00 | 1.45 | 98.55 | 0.00 | 0.55 | 99.45 | 0.00 | 2.05 | 95.42 | 2.53 | **0.727** | 2.859 | 1.725 | 1.620 | * |
| | | (0.1,0.5,0.5,0.5,0.6) | 2.89 | 97.11 | 0.00 | 0.46 | 99.54 | 0.00 | 1.71 | 98.29 | 0.00 | 0.34 | 99.66 | 0.00 | 1.87 | 95.57 | 2.56 | **0.742** | 2.840 | 1.680 | 1.840 | * |
| | (3,3,5,7,10) | (0.1,0.2,0.2,0.3,0.3) | 5.02 | 94.98 | 0.00 | 1.19 | 98.81 | 0.00 | 2.27 | 97.73 | 0.00 | 0.64 | 99.36 | 0.00 | 2.25 | 95.45 | 2.29 | **0.944** | 3.060 | 2.464 | 1.876 | * |
| | | (0.1,0.2,0.3,0.5,0.7) | 4.50 | 95.50 | 0.00 | 0.48 | 99.52 | 0.00 | 1.56 | 98.44 | 0.00 | 0.65 | 99.35 | 0.00 | 2.19 | 95.41 | 2.40 | **0.626** | 2.843 | 1.725 | 1.613 | * |
| | | (0.3,0.5,0.5,0.7,0.7) | 4.23 | 95.77 | 0.00 | 0.44 | 99.56 | 0.00 | 1.80 | 98.20 | 0.00 | 0.33 | 99.67 | 0.00 | 1.91 | 96.04 | 2.06 | **0.614** | 2.825 | 1.667 | 1.803 | * |

**Note**: Bold denotes the best-performing method.

* MOVER satisfies the CP criteria, and

** it is also the best-performing method.

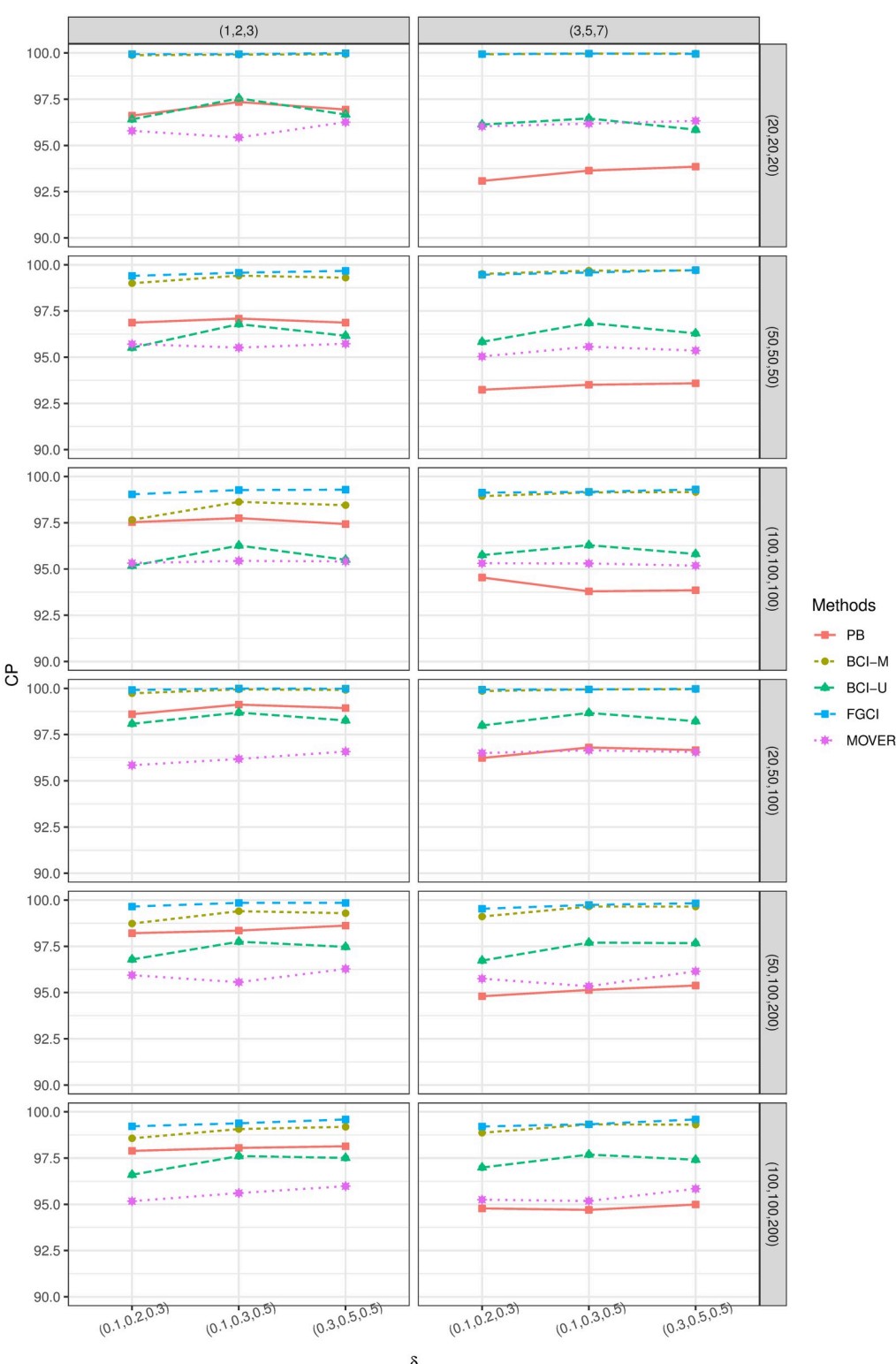

**Fig 1. CP performances of 95%SCIs for $\theta_{jl}$: 3 sample cases.**

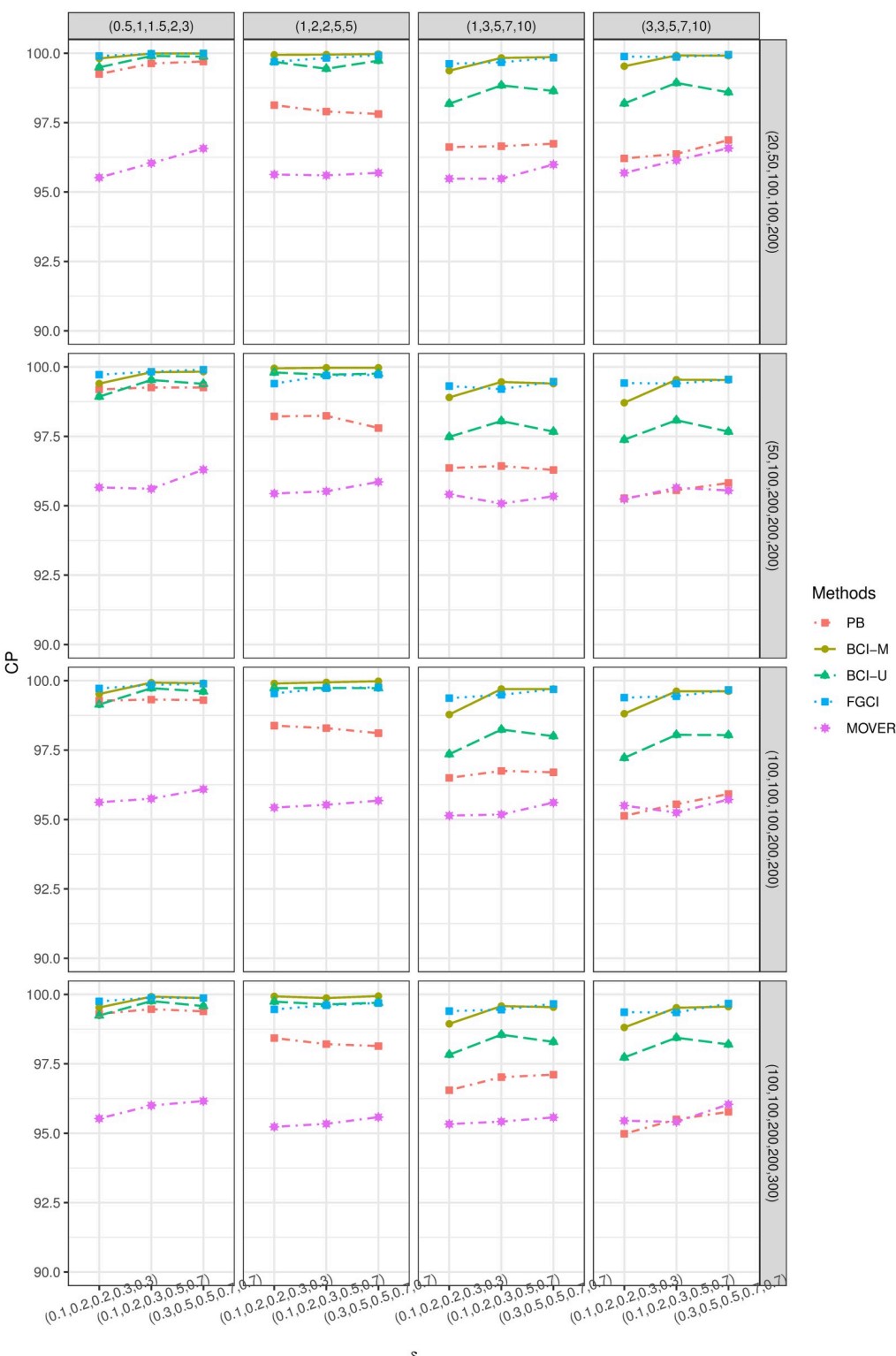

**Fig 2. CP performances of 95%SCIs for $\theta_{jl}$: 5 sample cases.**

**Table 3. Weekly natural rainfall records divided into five parts during July 2018.**

| Parts | Week | Weekly rainfall amounts (mm) | | | | | | | | | | | | | | | |
|---|---|---|---|---|---|---|---|---|---|---|---|---|---|---|---|---|---|
| Northern | 1 | 5 | 11.7 | 0 | 0 | 0 | 88.8 | 43 | 17.2 | 7.6 | 0 | 0 | 6.2 | 0 | 0 | 9.7 | 7.4 |
| | | 7.4 | 0 | 0 | 1.2 | 0 | 0 | 14.3 | 7.1 | 5.3 | 16.2 | 14.5 | 2.4 | 30 | 0 | 2.8 | 0 |
| | | 0 | 10 | 0 | 4.8 | 21 | 0 | 30 | 18.2 | 2.8 | 25.9 | 77 | 0 | 27.2 | 0 | 30.1 | 17.6 |
| | | 0 | 12.1 | 42.4 | 0 | 14.5 | 25.3 | 22.1 | 3.4 | 0 | 9.4 | 20.1 | 20.2 | 19.8 | 0 | | |
| | 2 | 4.1 | 12.2 | 0 | 0 | 0 | 19.1 | 43.7 | 17.9 | 18.3 | 0 | 0 | 10.3 | 0 | 0 | 16.4 | 36.8 |
| | | 24.2 | 0 | 0 | 16.5 | 0 | 0 | 23 | 54.1 | 84.4 | 97.1 | 21.5 | 62.9 | 102.1 | 4.8 | 0 | 6 |
| | | 0 | 0 | 0 | 0 | 18.1 | 3.2 | 1.8 | 0 | 12 | 37.9 | 128.3 | 0.9 | 61.4 | 0 | 12.5 | 5.5 |
| | | 0 | 2.2 | 14 | 0 | 12.2 | 54 | 50.5 | 30.3 | 0 | 0.7 | 49.5 | 24.3 | 39.7 | 0 | | |
| | 3 | 87.9 | 65.6 | 0 | 0 | 0 | 88.3 | 111.3 | 69.8 | 72.7 | 0 | 64.5 | 71.7 | 0 | 0 | 47.6 | 98.3 |
| | | 130.8 | 53.9 | 47.9 | 135.2 | 0 | 0 | 113 | 133.1 | 121.6 | 163.3 | 150.5 | 144 | 200.3 | 94.2 | 69.7 | 70 |
| | | 0 | 55.2 | 18 | 37.7 | 35.5 | 42.8 | 26.3 | 27.8 | 34.2 | 102 | 209.3 | 57 | 80.7 | 0 | 23.1 | 71.9 |
| | | 49.1 | 22.5 | 98.1 | 66.1 | 43.9 | 18.1 | 100.5 | 83 | 0 | 70.7 | 169.8 | 42.3 | 22.5 | 0 | | |
| | 4 | 141.8 | 65 | 0 | 0 | 0 | 46.1 | 136.2 | 39.6 | 60.3 | 0 | 25.1 | 58.7 | 0 | 0 | 32 | 21.5 |
| | | 57.6 | 22.7 | 0 | 29.5 | 0 | 0 | 104.5 | 90 | 97.1 | 120.7 | 121.4 | 56.4 | 119 | 39.2 | 58.8 | 10.3 |
| | | 0 | 25 | 0 | 0 | 21.6 | 1.6 | 23.2 | 0 | 40.1 | 108.8 | 245.4 | 11.2 | 93.4 | 0 | 34.4 | 34 |
| | | 41.2 | 15.5 | 86.1 | 51.9 | 43.3 | 2 | 65.7 | 42.8 | 0 | 68.4 | 36.8 | 8.9 | 81.5 | 0 | | |
| Central | 1 | 14.3 | 7.2 | 67.5 | 0 | 0 | 17 | 4.7 | 8.9 | 3.5 | 0 | 0 | 0 | 0 | 0 | 0 | 0 |
| | | 0 | 0 | 0 | 0 | 0 | 19.3 | 0 | 49.4 | 9.3 | 38.5 | 0 | 0 | 11.6 | 21 | 17.2 | 20.6 |
| | | 0 | 0 | 23.8 | 2.3 | 4.1 | 4.2 | 27.6 | 16.4 | 0 | 0 | 11.3 | 6.5 | 0 | 27.1 | 0 | 0 |
| | | 0 | 18.6 | 40.6 | 0 | 21 | 37.2 | 55.2 | 30.3 | 23.3 | | | | | | | |
| | 2 | 51.1 | 39.3 | 49.1 | 16.5 | 31.5 | 97.6 | 15.4 | 18 | 8.8 | 0 | 0 | 0 | 0 | 0 | 0 | 0 |
| | | 0 | 0 | 0 | 0 | 0 | 29.8 | 0 | 19.2 | 14.3 | 11.2 | 0 | 0 | 25.8 | 5.6 | 0 | 7.6 |
| | | 0 | 0 | 13.2 | 2 | 40.7 | 16 | 12.5 | 45.6 | 0 | 0 | 41.7 | 3 | 0 | 34.4 | 0 | 0 |
| | | 0 | 5.9 | 92 | 0 | 8.6 | 38.3 | 54.4 | 46.3 | 39 | | | | | | | |
| | 3 | 69.1 | 44.8 | 73.6 | 47.1 | 28 | 39.7 | 29.5 | 19.1 | 8.9 | 0 | 0 | 0 | 0 | 0 | 0 | 0 |
| | | 0 | 0 | 0 | 0 | 0 | 25.9 | 0 | 12.8 | 7.1 | 12.8 | 0 | 0 | 8.5 | 18.5 | 7 | 15.6 |
| | | 0 | 0 | 20.9 | 14 | 6.8 | 1.9 | 35.1 | 111 | 0 | 0 | 5.1 | 5.3 | 0 | 11.7 | 0 | 0 |
| | | 0 | 8.3 | 94.5 | 2.8 | 3.9 | 23.2 | 22 | 18.5 | 15 | | | | | | | |
| | 4 | 1.9 | 5.1 | 1 | 3.8 | 7 | 3.9 | 9 | 3.1 | 2.7 | 0 | 0 | 0 | 0 | 0 | 0 | 0 |
| | | 0 | 0 | 0 | 0 | 0 | 0 | 0 | 35.6 | 4.1 | 23.6 | 0 | 0 | 0 | 11 | 0 | 6.3 |
| | | 0 | 0 | 10.3 | 0.4 | 0.8 | 0.1 | 4.1 | 22.3 | 0 | 0 | 3.6 | 0 | 0 | 3.7 | 0 | 0 |
| | | 0 | 0 | 229.2 | 0 | 1.1 | 42.7 | 50.3 | 54.6 | 16.4 | | | | | | | |
| East | 1 | 23.4 | 36.1 | 13.4 | 1.5 | 12.6 | 19.2 | 10 | 4.8 | 15.8 | 36 | 6.5 | 22.4 | 47.1 | 36.8 | 13.6 | 61 |
| | | 26.3 | 7.5 | 102.5 | 20.4 | 24.9 | 14.6 | 87.5 | 306.9 | 72.8 | 168.3 | 89.5 | 78.6 | 26.5 | | | |
| | 2 | 106 | 49.4 | 118.9 | 117.4 | 110.4 | 9.3 | 13.2 | 1.5 | 33.7 | 33.2 | 4.1 | 23.1 | 48.7 | 81.5 | 24.3 | 70.7 |
| | | 30.1 | 34.4 | 154.1 | 45.6 | 119.1 | 42 | 92.9 | 193.6 | 198.9 | 204.9 | 58.1 | 114.5 | 16.8 | | | |
| | 3 | 33.5 | 32.9 | 34 | 70 | 45.3 | 4.9 | 8 | 3.3 | 14.8 | 4.3 | 8.4 | 5.5 | 45.7 | 189.5 | 10.8 | 78.9 |
| | | 153.9 | 92.5 | 100.5 | 12.5 | 50 | 23.9 | 316.9 | 585.5 | 248.8 | 113.5 | 79.6 | 253 | 131.9 | | | |
| | 4 | 59.1 | 21.8 | 196.4 | 174.6 | 74.5 | 0.3 | 0 | 0 | 0 | 0 | 0.1 | 0.3 | 8.7 | 2.7 | 0 | 10 |
| | | 2 | 0 | 246.1 | 49.9 | 31.2 | 2.6 | 23 | 19.8 | 5 | 434.9 | 0 | 20.6 | 0.5 | | | |

*(Continued)*

**Table 3.** (Continued)

| Parts | Week | Weekly rainfall amounts (mm) | | | | | | | | | | | | | | | |
|---|---|---|---|---|---|---|---|---|---|---|---|---|---|---|---|---|---|
| Southeast | 1 | 19.6 | 0 | 9.8 | 22.6 | 4.6 | 50.6 | 53 | 27.5 | 37.5 | 129.6 | 6 | 34.2 | 52.7 | 65.3 | 59.8 | 47.5 |
| | | 45.5 | 50 | 69 | 34.9 | 7.3 | 5 | 33.4 | 17.9 | 4 | 13 | 21 | 0 | 20.2 | 6.8 | 0 | 32.9 |
| | | 20 | 45.8 | 16.1 | 20.7 | 0 | 31.2 | 71.5 | 13.4 | 2.5 | 0 | 12.4 | 39.8 | 27.6 | 21.8 | 18 | 1 |
| | | 0.5 | 0 | 8 | 20 | 1 | 10 | 12.8 | 0 | 4.7 | 0 | 0 | 70.9 | 26.5 | 12.6 | 35 | 6.8 |
| | | 3.9 | 30.9 | 0.5 | 15.1 | 35 | 0 | 8 | 9.3 | 26.5 | 0 | 23.5 | 0 | 46.1 | 25.6 | 61.3 | 13.4 |
| | | 52 | 86.5 | 21.6 | 3.2 | 25.5 | 19.5 | 15.2 | 0 | 68.6 | | | | | | | |
| | 2 | 23.5 | 0 | 34.7 | 2.9 | 12.3 | 115.3 | 116.8 | 182.5 | 92.6 | 319.9 | 155 | 123 | 240 | 52.3 | 40.3 | 10 |
| | | 24.7 | 0.9 | 98.5 | 10.9 | 12.1 | 4.8 | 21.3 | 5 | 0 | 14 | 4 | 19.5 | 54.2 | 22.8 | 0 | 11.6 |
| | | 72 | 27.4 | 18.2 | 20.5 | 0.5 | 47.2 | 30.4 | 6.2 | 3.3 | 0 | 13.7 | 0.4 | 13.6 | 4.8 | 0 | 0 |
| | | 0 | 16 | 0 | 0 | 0 | 0 | 0 | 0 | 0 | 5.2 | 0 | 0 | 0 | 0 | 0 | 0 |
| | | 0 | 0 | 0 | 0 | 0 | 0 | 0 | 0 | 0 | 0 | 0.5 | 0 | 0 | 0 | 0 | 0 |
| | | 6.9 | 3.3 | 0 | 0 | 0 | 0 | 0 | 0 | 0 | | | | | | | |
| | 3 | 21.8 | 0 | 60.8 | 33.7 | 34 | 67.2 | 88.2 | 53.2 | 75 | 233.4 | 76.1 | 50.4 | 132 | 25.8 | 24.9 | 12.8 |
| | | 74 | 0 | 20 | 22.2 | 12.3 | 40.2 | 29.9 | 6.2 | 9 | 16.2 | 0 | 25.5 | 63.5 | 22.9 | 2 | 30.4 |
| | | 30 | 37.2 | 29.8 | 31.6 | 18.7 | 50.9 | 43.9 | 10.2 | 24.3 | 0 | 26.8 | 15.1 | 25.8 | 17.8 | 5.3 | 0 |
| | | 9 | 40.7 | 9 | 0 | 0 | 8 | 22.1 | 25 | 5 | 0 | 0 | 7 | 12.1 | 7.5 | 11.3 | 7.5 |
| | | 5.1 | 4.7 | 6.4 | 0.5 | 1.5 | 3.1 | 0 | 0 | 6.2 | 6.4 | 0 | 0 | 14.4 | 0 | 2.7 | 1.5 |
| | | 0 | 17.7 | 0 | 0 | 4.5 | 0 | 7.1 | 0 | 0 | | | | | | | |
| | 4 | 3.4 | 0 | 13 | 0.5 | 1.6 | 26.2 | 22 | 59.8 | 11 | 232.4 | 18.3 | 27.1 | 82.5 | 7.1 | 0.2 | 1.9 |
| | | 34.6 | 0 | 88 | 1.7 | 1.7 | 0.5 | 21 | 18.5 | 0 | 0 | 0 | 0 | 30 | 3.5 | 0 | 0 |
| | | 25 | 55.5 | 9.3 | 23 | 0 | 66.7 | 40.2 | 6.9 | 5.5 | 0 | 0.3 | 15.4 | 3.4 | 30.8 | 0 | 6 |
| | | 62.3 | 21.2 | 0 | 0 | 0 | 0 | 3.9 | 3 | 0 | 0 | 0 | 31 | 146.5 | 45.4 | 25.5 | 67.3 |
| | | 34.6 | 65.7 | 107.6 | 11 | 89.6 | 32.6 | 0 | 116.3 | 104.2 | 59.4 | 0 | 0 | 47.7 | 85.2 | 250.7 | 112.4 |
| | | 67.6 | 198.8 | 85.2 | 83.4 | 117.6 | 46.5 | 120.9 | 0 | 111.9 | | | | | | | |
| Southwest | 1 | 234.5 | 0 | 0 | 0 | 154.7 | 0 | 0 | 0 | 0 | 45 | 57.1 | 18.6 | 0 | 17.5 | 0 | 0 |
| | | 83.5 | 0 | 0 | 0 | 0 | 0 | 0 | 0 | 0 | 0 | 0 | 78.6 | 0 | 29.5 | | |
| | 2 | 334.3 | 0 | 0 | 0 | 94.8 | 0 | 0 | 0 | 0 | 26 | 52.5 | 34 | 0 | 27.7 | 0 | 0 |
| | | 95.5 | 0 | 0 | 0 | 0 | 0 | 0 | 0 | 0 | 0 | 0 | 102.8 | 0 | 74 | | |
| | 3 | 147.7 | 0 | 0 | 0 | 139.9 | 0 | 0 | 0 | 0 | 44.8 | 26.5 | 0 | 0 | 34.6 | 0 | 0 |
| | | 103 | 0 | 0 | 0 | 0 | 0 | 0 | 0 | 0 | 0 | 0 | 65.8 | 26.3 | 45.9 | | |
| | 4 | 262.9 | 0 | 0 | 0 | 43 | 0 | 0 | 0 | 0 | 12.6 | 6.3 | 0 | 0 | 56.2 | 0 | 0 |
| | | 37.5 | 0 | 0 | 0 | 0 | 0 | 0 | 0 | 0 | 0 | 0 | 131.9 | 0 | 53.7 | | |

Source: Thailand Meteorological Department. URL: https://www.tmd.go.th/services/weekly_report.php

are four weeks in July 2018: week 1 (2–8), week 2 (9–15), week 3 (16–22), and week 4 (23–29). There were zero observations (i.e., no rainfall) at all of the substations. Histogram plots (Fig 3) and normal Q-Q plots (Fig 4) show that the datasets from almost all areas were consistent with the assumptions of delta-lognormality (the northeast area was omitted as it did not fill the requirements). Furthermore, AIC can be used to check the fit of a particular distribution [34]. It is defined as

$$\text{AIC} = -2\ln L + 2p \tag{25}$$

where $L$ is the likelihood function and $p$ is the number of parameters in the model. To judge a suitable distribution for the data, it is considered from one that has minimum AIC. From the AIC results (Table 4), it indicates that the positive rainfall observations fit the lognormal

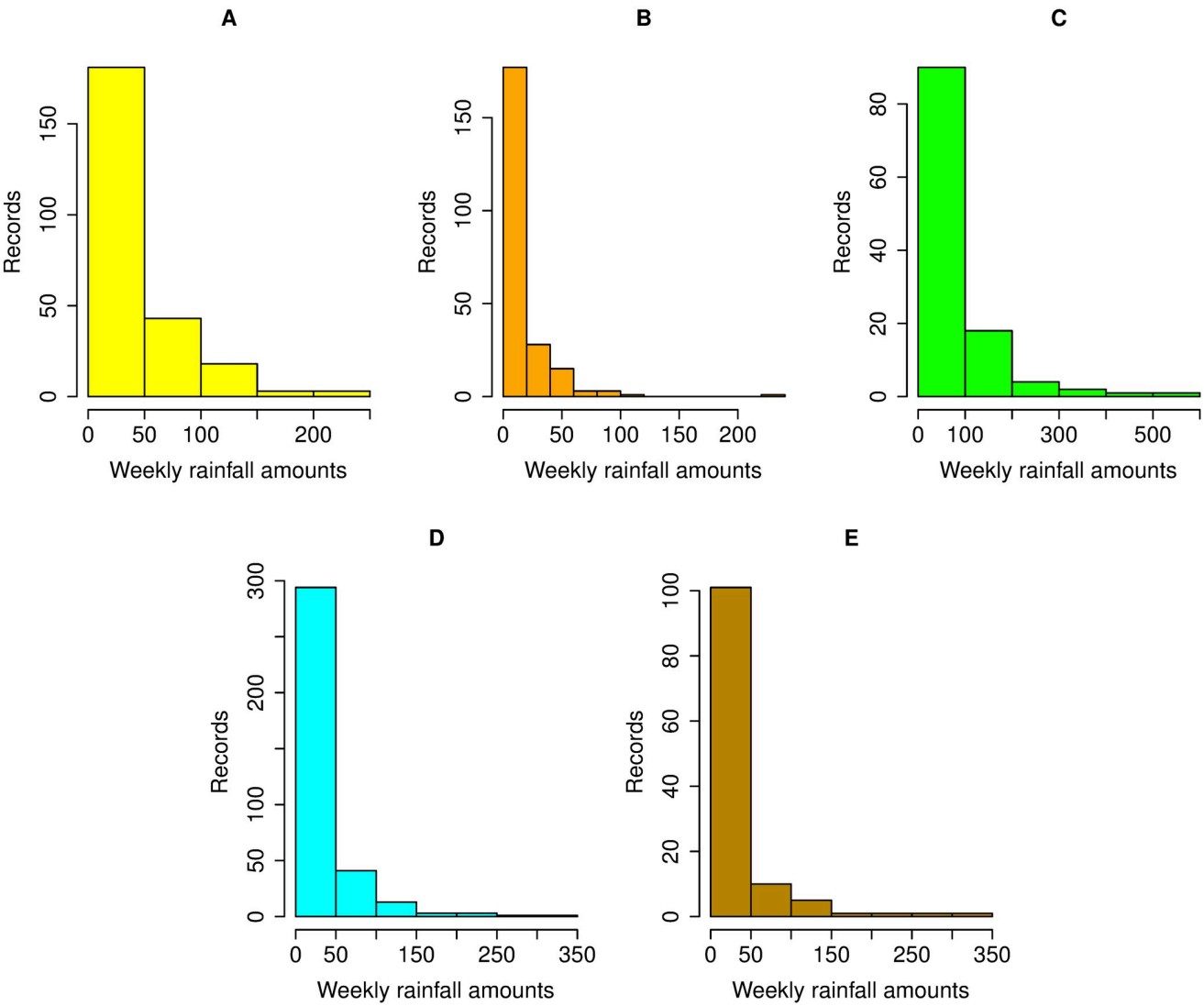

**Fig 3. Histogram plots of weekly rainfall records in five parts.** (A) Northern (B) Central (C) Eastern (D) Southeastern (E) Southwestern.

distribution for all of the study areas. Likewise, these datasets also contained zero observations, so we conclude that the data for five areas follow the assumptions of delta-lognormal distributions. The data summary is reported in Table 5.

The 95% SCIs for $\theta_j; j = 1, 2, 3, 4, 5$, were computed to estimate pairwise differences in the means of the weekly rainfall datasets for the five areas in Thailand (Table 6). For sample case $k = 5$, it can be seen that MOVER had more efficient (narrower) intervals than the other methods for small differences in variance $\sigma_j^2$ and proportion of zero $\delta_j$ and large unequal sample sizes, which is in line with the simulation results in the previous section.

## Discussion

We conducted simultaneous pairwise comparisons of the means of delta-lognormal distributions in a simulation study and five datasets containing zero observations and where the

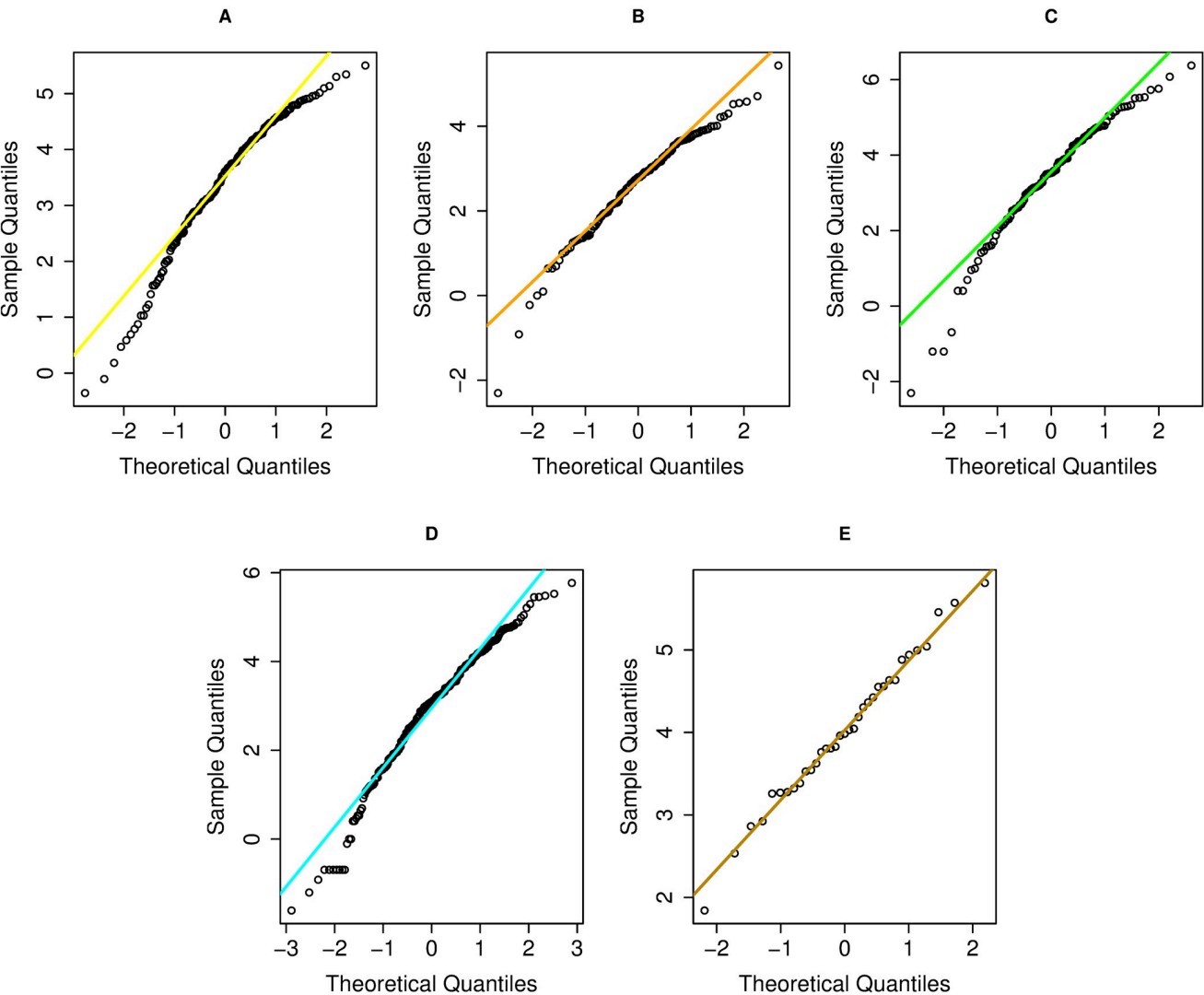

**Fig 4. Normal Q-Q plots of log-transformation of non-zero records in five parts.** (A) Northern (B) Central (C) Eastern (D) Southeastern (E) Southwestern.

positive observations were right-skewed using SCIs based on PB, FGCI, MOVER, and Bayesian credible intervals (BCI-M and BCI-U).

The simulation results provided insight into the analytical behavior of the SCIs. For small sample cases ($k = 3$), it was found that BCI-U is an appropriate method for small-to-large

**Table 4. AIC results for nonzero rainfall amounts in five parts.**

| Distribution | AIC values | | | | |
|---|---|---|---|---|---|
| | **Northern** | **Central** | **Eastern** | **Southeastern** | **Southwestern** |
| Cauchy | 1859.046 | 1131.425 | 1233.443 | 2568.220 | 393.113 |
| Logistic | 1831.410 | 1144.392 | 1266.594 | 2636.477 | 397.227 |
| Lognormal | **1753.656** | **1057.969** | **1151.703** | **2401.707** | **374.674** |
| Normal | 1844.458 | 1195.499 | 1300.466 | 2724.489 | 403.431 |
| T-distribution | 1830.311 | 1121.858 | 1234.366 | 2559.868 | 393.281 |

**Table 5. Estimated parameters for five parts in Thailand.**

| Estimates | Thailand parts | | | | |
|---|---|---|---|---|---|
| | **Northern** | **Central** | **Eastern** | **Southeastern** | **Southwestern** |
| $n$ | 248 | 228 | 116 | 356 | 120 |
| $\hat{w}$ | 3.40 | 2.61 | 3.38 | 2.89 | 4.03 |
| $s^2$ | 1.37 | 1.46 | 2.56 | 1.91 | 0.76 |
| $\hat{\delta}$ | 0.29 | 0.45 | 0.06 | 0.27 | 0.71 |
| $\hat{\theta}$ | 41.81 | 15.35 | 96.71 | 33.67 | 23.60 |

differences in $\sigma_j^2$ and $\delta_i$ together with equal sample sizes due to the posterior distribution of $\sigma_j^2$ based on the uniform prior being closer to the real $\sigma_j^2$ than the mixed prior. When the differences in $\sigma_j^2$ were small, MOVER was the next best method, the reason being that the CI of $\sigma_j^2$ is limited for cases of large differences of $\sigma_j^2$. On the other hand, PB could handle this problem because it uses data resampling with replacement, which is a different approach from the other methods. For large sample cases ($k = 5$), PB performance was stable and similar to the small sample case. For small differences in $\sigma_j^2$, MOVER was good for small differences in $\delta_i$ because it is affected by the CI for $\delta_i'$ based on variance stabilized transformation in Eq 11, whereas PB is suitable for large differences in $\delta_i$. This means that the bootstrap resampling could solve the limitation of applying MOVER.

Importantly, the practical application of these methods was demonstrated by estimating the pairwise differences between the mean of the natural rainfall datasets for the five areas in Thailand. Comparisons between the areas indicate that the weekly rainfall mean in the eastern area was greater than the others. The reason for this could be that the eastern area coastline is on the Gulf of Thailand where there are frequent heavy rainstorms. Importantly, the occurrences

**Table 6. 95%SCIs for all differences of weekly rainfall means.**

| Comparison | Mean difference | PB | | | FGCI | | | MOVER | | | BCI-M | | | BCI-U | | |
|---|---|---|---|---|---|---|---|---|---|---|---|---|---|---|---|---|
| | | **L** | **U** | **Length** | **L** | **U** | **Length** | **L** | **U** | **Length** | **L** | **U** | **Length** | **L** | **U** | **Length** |
| Northern/ Central | 26.46 | 10.82 | 42.09 | 31.27 | 9.12 | 43.80 | 34.68 | 16.10 | 39.44 | **23.35** | -38.97 | 91.88 | 130.85 | -41.68 | 90.22 | 131.90 |
| Northern/ Eastern | -54.90 | -120.48 | 10.67 | 131.15 | -127.62 | 17.82 | 145.44 | -126.13 | -21.34 | **104.80** | -120.33 | 10.52 | 130.85 | -120.85 | 11.04 | 131.90 |
| Northern/ Southeastern | 8.14 | -10.20 | 26.48 | 36.68 | -12.20 | 28.48 | 40.68 | -5.12 | 22.30 | **27.42** | -57.29 | 73.56 | 130.85 | -57.81 | 74.09 | 131.90 |
| Northern/ Southwestern | 18.21 | -2.32 | 38.73 | 41.05 | -4.55 | 40.97 | 45.52 | 1.24 | 33.12 | **31.88** | -47.22 | 83.63 | 130.85 | -47.74 | 84.16 | 131.90 |
| Central/Eastern | -81.36 | -145.73 | -16.99 | 128.74 | -152.74 | -9.98 | 142.76 | -152.20 | -49.58 | **102.62** | -146.79 | -15.94 | 130.85 | -147.31 | -15.41 | 131.90 |
| Central/ Southeastern | -18.32 | -31.72 | -4.92 | 26.80 | -33.18 | -3.46 | 29.72 | -29.28 | -9.18 | **20.11** | -83.74 | 47.11 | 130.85 | -84.27 | 47.63 | 131.90 |
| Central/ Southwestern | -8.25 | -24.51 | 8.02 | 32.53 | -26.28 | 9.79 | 36.07 | -23.49 | 2.02 | **25.51** | -73.67 | 57.18 | 130.85 | -74.20 | 57.70 | 131.90 |
| Eastern/ Southeastern | 63.04 | -2.04 | 128.12 | 130.16 | -9.13 | 135.21 | 144.34 | 30.20 | 134.11 | **103.90** | -2.38 | 128.47 | 130.85 | -2.91 | 128.99 | 131.90 |
| Eastern/ Southwestern | 73.11 | 7.38 | 138.84 | 131.46 | 0.22 | 146.00 | 145.78 | 38.61 | 144.33 | **105.72** | 7.69 | 138.54 | 130.85 | 7.16 | 139.06 | 131.90 |
| Southeastern/ Southwestern | 10.07 | -8.81 | 28.95 | 37.76 | -10.86 | 31.01 | 41.87 | -6.19 | 23.26 | **29.44** | -55.35 | 75.50 | 130.85 | -55.88 | 76.02 | 131.90 |

of extreme rainfall in the northern area can be used to generate warning signals of imminent natural disasters (flooding and landslides) for notifying people and preparing them in advance. The results of this empirical application were similar to the simulation study results in that MOVER was appropriate for small differences in $\sigma_j^2$ and $\delta_j$ together with large unequal sample sizes due to the lower and upper limits covering all pairwise differences in the delta-lognormal means of the datasets together with having the narrowest lengths.

## Conclusion

The objective of this study was to use SCIs to analyze the pairwise differences between the means of multiple delta-lognormal distributions. Derivations of the proposed methods for constructing the SCIs: PB, FGCI, MOVER, BCI-M, and BCI-U were provided. Their performances were evaluated via simulation studies and an empirical application. From the results, BCI-U and PB are the recommended methods for equal and unequal sample sizes, respectively, with large differences in $\sigma_j^2$. The next best method was MOVER for small differences in $\sigma_j^2$. For large sample cases ($k = 5$), MOVER is also recommended for small differences in $\sigma_j^2$ and $\delta_i$. PB can also be recommended for large differences in $\sigma_j^2$. Meanwhile, both BCI-U and FGCI are good alternatives for $k = 5$.

## Supporting information

**S1 Abbreviations. Abbreviations commonly used throughout this article.**
(PDF)

**S1 Appendix. Theorems and proofs of methods to formulate the $100(1 − \alpha)$% simultaneous confidence intervals (SCIs) for $\theta_{ij}$.**
(PDF)

**S1 Code. The R code for computing the performances of the proposed methods as $100(1 − \alpha)$%SCIs for $\theta_{ij}$.**
(R)

**S1 Data. Weekly natural rainfall records during July 2018 for the five areas of Thailand used with R code.**
(XLSX)

**S1 Fig. CP performances of 95%SCIs for $\theta_{jl}$: 3 sample cases.**
(PDF)

**S2 Fig. CP performances of 95%SCIs for $\theta_{jl}$: 5 sample cases.**
(PDF)

**S3 Fig. Histogram plots of the weekly rainfall records for the five areas of Thailand.** (A) Northern, (B) Central, (C) Eastern, (D) Southeastern, and (E) Southwestern.
(PDF)

**S4 Fig. Normal Q-Q plots of the log-transformed non-zero records for the five areas of Thailand.** (A) Northern (B) Central (C) Eastern (D) Southeastern (E) Southwestern.
(PDF)

**S1 Table. Performance metrics for the 95% SCIs for $\theta_{jl}$: 3 sample cases.**
(PDF)

**S2 Table. Performance metrics for the 95% SCIs for $\theta_{jl}$: 5 sample cases.**
(PDF)

**S3 Table. Weekly natural rainfall records for the five areas of Thailand during July 2018.**
(PDF)

**S4 Table. AIC results for non-zero rainfall amounts in the five areas of Thailand.**
(PDF)

**S5 Table. Estimated statistical parameters for the five areas of Thailand.**
(PDF)

**S6 Table. The 95% SCIs for all pairwise differences between the means of the weekly rainfall data for the five areas of Thailand.**
(PDF)

## Acknowledgments

The authors are grateful to the Academic Editor and the Reviewers for their valuable comments and suggestions which help to improve this manuscripts.

## Author Contributions

**Conceptualization:** Sa-Aat Niwitpong, Suparat Niwitpong.

**Data curation:** Patcharee Maneerat.

**Formal analysis:** Patcharee Maneerat, Sa-Aat Niwitpong.

**Funding acquisition:** Suparat Niwitpong.

**Investigation:** Patcharee Maneerat, Sa-Aat Niwitpong.

**Methodology:** Patcharee Maneerat, Sa-Aat Niwitpong, Suparat Niwitpong.

**Project administration:** Suparat Niwitpong.

**Resources:** Patcharee Maneerat.

**Software:** Patcharee Maneerat.

**Supervision:** Sa-Aat Niwitpong, Suparat Niwitpong.

**Validation:** Patcharee Maneerat.

**Visualization:** Sa-Aat Niwitpong.

**Writing – original draft:** Patcharee Maneerat.

**Writing – review & editing:** Suparat Niwitpong.

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
