## [Decision Letter · Decision Letter 0]

9 Feb 2021

PONE-D-20-38684

Simultaneous confidence intervals for all pairwise comparisons of the means of delta-lognormal distributions with application to rainfall data

PLOS ONE

Dear Dr. Niwitpong,

Thank you for submitting your manuscript to PLOS ONE. After careful consideration, we feel that it has merit but does not fully meet PLOS ONE’s publication criteria as it currently stands. Therefore, we invite you to submit a revised version of the manuscript that addresses the points raised during the review process.

We look forward to receiving your revised manuscript.

Kind regards,

Mohammad Asghari Jafarabadi

Academic Editor

PLOS ONE

2. Please ensure that you refer to Figures 1-4 in your text as, if accepted, production will need this reference to link the reader to the figures.

3. Please upload a copy of Figures 1-4, to which you refer in your text on page 22. If the figure is no longer to be included as part of the submission please remove all reference to it within the text.

4. We note you have included a table to which you do not refer in the text of your manuscript. Please ensure that you refer to Table 5 in your text; if accepted, production will need this reference to link the reader to the Table.

Reviewers' comments:

Reviewer's Responses to Questions

**Comments to the Author**

1. Is the manuscript technically sound, and do the data support the conclusions?

Reviewer #1: Yes

Reviewer #2: Yes

Reviewer #3: No

2. Has the statistical analysis been performed appropriately and rigorously? 

Reviewer #1: Yes

Reviewer #2: Yes

Reviewer #3: No

3. Have the authors made all data underlying the findings in their manuscript fully available?

Reviewer #1: Yes

Reviewer #2: Yes

Reviewer #3: Yes

4. Is the manuscript presented in an intelligible fashion and written in standard English?

Reviewer #1: Yes

Reviewer #2: Yes

Reviewer #3: No

5. Review Comments to the Author

Reviewer #1: The authors present a good study on the topic meeting all the necessary requirements of a new research. So I recommend the publication of the manuscript in its present form in PLOS ONE. However, the following correction may be made in the revised draft.

In line 59, 'is' should be replaced with 'as'.

Reviewer #2: Thank you for giving me the opportunity to review this manuscript. It is a solid and interesting paper, however, the focus is very much on mathematical aspects/proofs of statistical techniques that had not yet been fully developed for the special case of the delta-lognormal distribution. I am actually surprised that the authors didn't choose a statistical journal. Also the majority of references stem from explicitly statistical journals.

As the focus of PLOS ONE is decidedly interdisciplinary, I think this paper in its current form is far too technical to be of use to anyone outside the statistics/math community. In this form it would probably be better suited for a statistical journal. The motivation for the methods developed in this paper is rainfall data, but I think that most people working in that area would have considerable difficulties in understanding that paper and not find it useful for their work.

Therefore, I recommend a thorough rewrite of the paper to be much more understandable for readers from various disciplines before it can be published on PLOS ONE. Please finde my specific comments below:

Major points:

1. Rewrite the manuscript as a "story" for non-statisticians. This includes

- Move theorems and proofs in an appendix where people with a specific mathematical interest can find them.

- Explain to non-statisticians why the issue of having simultaneous confidence bands is important.

- Formulate with word what you are doing mathematically and why you are doing that

- Stress even more the practical use of your methods, especially in introduction and discussion.

- You use a lot of abbreviations. This is necessary to avoid to lengthen the script, but it also makes it even harder to read. Maybe you could add a list of all abbreviations to look them up if necessary?

- I was surprised to see that the word "sampling" or "sampling distribution" never occurs in that paper although many methods are sampling-based. Please change that and give more explanations of the differences between the methods for non-statisticians.

- To make your proposed methods useable in practice, you should not only explain the algorithms, but maybe provide the code or at least give information on statistical programs to use.

- Take more time to explain the results of both the simulations and the application and their respective meanings.

2. I don't understand the relative average lengths. Why is MOVER taken as reference? What is meant with "less than 1 and the minimal value"? What is this minimal value? Would not a ratio smaller than 1 mean that the respective SCI is better (given a comparable coverage probability)? Because that would mean it's smaller than the corresponding MOVER SCI? I have probably understood something wrong, but this should be all the more a reason to explain that better.

3. The authors use AIC on page 14 and in Table 4, but they don't explain how they get it. Give a thorough explanation how you provide those values and how the models are exactly calculated (apart from the different underlying distributions).

4. Please mention which statistical program you used and if possible provide your code.

Minor points:

1. Would it be possible to also provide the data set in the supplementary material in a form ready to be used? If one wanted to use the data in the current form one would have to type all values from Table 3 in a data set.

2. Sometimes definitions are hard to find or not there. For example, T on page 4 and R on page 6 seem to come out of nowhere.

3. I couldn't find where the abbreviation "CLT" on page 9 is introduced. As said above, a list of all the abbreviations would be helpful.

4. Delete the information on the flooded cave system from p.14. It has already been given in the introduction and is of no additional use here.

5. Mention what bold face means in the tables. I couldn't find that information.

6. What does "MOVER satisfies the accurate criteria" (tables) mean?

Reviewer #3: This paper builds upon several works on simultaneous confidence intervals (SCI) based on parametric bootstrap, fiducial confidence interval, variance estimates recovery and credible Bayesian intervals. The latter based on mixed and uniform priors. The authors focus on the delta-lognormal distribution and motivate their study by claiming that SCI can help decision-makers to avoid the impact of unexpected events during the rainy season.

See File in the attachment for the complete review.

6. PLOS authors have the option to publish the peer review history of their article (what does this mean?). If published, this will include your full peer review and any attached files.

Reviewer #1: No

Reviewer #2: **Yes: **Julia Braun

Reviewer #3: **Yes: **Antonino Abbruzzo

---

## [Author Response · Author response to Decision Letter 0]

30 Mar 2021

Response to Academic Editor and Reviewers

Journal: PLOS One

Manuscript ID: PONE-D-20-38684

Title name: Simultaneous confidence intervals for all pairwise comparisons of the means of delta-lognormal distributions with application to rainfall data

Authors: Patcharee Maneerat, Sa-Aat Niwitpong and Suparat Niwitpong

Response to Academic Editor 

Dear Academic Editor: 

We would like to thank you for your time in reviewing our manuscript. The following are our responses to each of your comments.

Academic Editor: 

Response: Thank you for your suggestions. We have subsequently formatted our manuscript to follow the PLOS ONE style requirements.

2. Please ensure that you refer to Figures 1-4 in your text as, if accepted, production will need this reference to link the reader to the figures.

Response: We have ensured that Figures 1-4 are cited in the text of our manuscript.

3. Please upload a copy of Figures 1-4, to which you refer in your text on page 22. If the figure is no longer to be included as part of the submission please remove all reference to it within the text.

Response: Figures 1-4 have been uploaded with our manuscript.

4. We note you have included a table to which you do not refer in the text of your manuscript. Please ensure that you refer to Table 5 in your text; if accepted, production will need this reference to link the reader to the Table.

Response: Line 290, Table 5 has been cited in the text and included in the manuscript.

Response: We have included captions in the Supporting Information files and updated the citations accordingly.

Dear Reviewers: 

We would like to thank you for your time and patience in reviewing our manuscript and making insightful comments toward improving it for resubmission. The following are our responses to each of your comments.

Reviewers' comments:

 Reviewer #1: 

The authors present a good study on the topic meeting all the necessary requirements of a new research. So I recommend the publication of the manuscript in its present form in PLOS ONE. However, the following correction may be made in the revised draft. In line 59, 'is' should be replaced with 'as'.

Response: Thanks for your valuable comments. In Line 73, we have replaced “is” with “as”.

Reviewer #2: 

Major points:

1. Rewrite the manuscript as a "story" for non-statisticians. This includes

1.1 Move theorems and proofs in an appendix where people with a specific mathematical interest can find them.

Response: Thanks for this comment. We have moved the theorems and proofs to S1 Appendix.

1.2 Explain to non-statisticians why the issue of having simultaneous confidence bands is important.

Response: In Paragraph 4 of the Introduction, we have provided reasons for formulating simultaneous confidence intervals (SCIs) for the means of delta-lognormal distributions and illustrating their use on the natural rainfall data from five areas in Thailand.

1.3 Formulate with word what you are doing mathematically and why you are doing that

- Stress even more the practical use of your methods, especially in introduction and discussion.

Response: Thanks for your suggestions. We have attempted to do so in Paragraph 3 the Introduction and Paragraph 1 of the Discussion, as well as in the Methods by explaining the mathematical approaches in more detail.

1.4 You use a lot of abbreviations. This is necessary to avoid to lengthen the script, but it also makes it even harder to read. Maybe you could add a list of all abbreviations to look them up if necessary?

Response: Abbreviations used throughout this article have been added to Supporting Information (S1 Abbreviations). 

1.5 I was surprised to see that the word "sampling" or "sampling distribution" never occurs in that paper although many methods are sampling-based. Please change that and give more explanations of the differences between the methods for non-statisticians.

Response: Thank you for your concerns. To the best of our knowledge, the word “sampling” and related phrases is a technical term in the field of statistics. For example, bootstrap sampling is a technique that involves drawing repeated samples (resampling with replacement) from the original data. (Efron B.and Tibshirani R. J., 1993). In Paragraph 2 in the Discussion section, we have explained the differences between the methods for non-statisticians.

Reference: 

Efron B, Tibshirani RJ. An Introduction to the Bootstrap. Chapman & Hall /CRC; 1993.

1.6 To make your proposed methods useable in practice, you should not only explain the algorithms, but maybe provide the code or at least give information on statistical programs to use.

Response: Thank you for pointing this out. In Paragraph 2 of the Simulation Studies section, we have provided information on the statistical computing used to evaluate the SCI performances.

1.7 Take more time to explain the results of both the simulations and the application and their respective meanings.

Response: In Paragraph 2 of An Example using Real Data and Paragraph 3 of Discussion, we have carefully explained the empirical application associated with the simulation results.

2. I don't understand the relative average lengths. Why is MOVER taken as reference? What is meant with "less than 1 and the minimal value"? What is this minimal value? Would not a ratio smaller than 1 mean that the respective SCI is better (given a comparable coverage probability)? Because that would mean it's smaller than the corresponding MOVER SCI? I have probably understood something wrong, but this should be all the more a reason to explain that better.

Response: In Paragraph 1 of the Simulation Studies section, we have added a straightforward explanation of the relative average length (RAL) for easy understanding. A method with an RAL of less than 1 has a shorter average length (and thus performed better) than the one it is being compared against. Moreover, we have provided the reason why MOVER was chosen as the reference. 

3. The authors use AIC on page 14 and in Table 4, but they don't explain how they get it. Give a thorough explanation how you provide those values and how the models are exactly calculated (apart from the different underlying distributions).

Response: In Paragraph 1 of An Example using Real Data, we have explained the use the AIC to investigate which distribution provides the best fit of the data.

4. Please mention which statistical program you used and if possible provide your code.

Response: As mentioned in Paragraph 2 in the Simulation Studies section, we used R studio for the statistical computing, and we added the R code to the Supporting Information.

Minor points:

1. Would it be possible to also provide the data set in the supplementary material in a form ready to be used? If one wanted to use the data in the current form one would have to type all values from Table 3 in a data set.

Response: As requested, we have provided the data including the R code file in the Supporting Information.

2. Sometimes definitions are hard to find or not there. For example, T on page 4 and R on page 6 seem to come out of nowhere.

Response: On page 4, “T^((PB))” stands for the parametric bootstrap variable, while “R” denotes the fiducial generalized pivotal quantity on page 5. 

3. I couldn't find where the abbreviation "CLT" on page 9 is introduced. As said above, a list of all the abbreviations would be helpful.

Response: The abbreviation "CLT" has been defined and also included in the Supporting Information (S1 Abbreviations). 

4. Delete the information on the flooded cave system from p.14. It has already been given in the introduction and is of no additional use here.

Response: Thanks for your suggestions. In Paragraph 1 of An Example using Real Data, it was deleted.

5. Mention what bold face means in the tables. I couldn't find that information.

Response: Tables 1-2, We added the meaning of bold face in table footnotes.

6. What does "MOVER satisfies the accurate criteria" (tables) mean?

Response: Thank you for your comment. We meant that MOVER’s performance was the best under those circumstances. However, we have removed this misleading text from the manuscript. 

Reviewer #3:

1.This paper builds upon several works on simultaneous confidence intervals (SCI) based on parametric bootstrap, fiducial confidence interval, variance estimates recovery and credible Bayesian intervals. The latter based on mixed and uniform priors. The authors focus on the delta-lognormal distribution and motivate their study by claiming that SCI can help decision-makers to avoid the impact of unexpected events during the rainy season. The problem of simultaneous confidence intervals is interesting, and extreme events are both challenging and of extreme importance. However, I don’t think the simultaneous confidence interval for delta log-normal would help the decision-makers avoid unexpected events during the rainy season. 

Response: Thank you for this important criticism. We think that the simultaneous confidence interval could provide important information that, in conjunction with other related information, can be used to make decisions that will help to prepare for extreme events in advance. 

2. Moreover, this paper is hard to follow due to some involving notation, that the authors should simplify. 

Response: Thank you for pointing this out. We moved all of the theorems to Supporting Information (S1 Appendix) where a reader with a specific interest in the mathematics can find them.

3. The results look simple extension of the following papers [2], [1], [3] and [4]. The authors must clarify their contribution. For example, it is not clear whether the authors propose the theorems presented in the paper. 

Response: This is an excellent point raised by the reviewer. In Paragraph 4 of the Introduction, we have clarified our contribution and clarified its context in relation to studies [1–4]. In Supporting information (S1 Appendix), we have carefully checked the contexts of the theorems, as the reviewer requested.

References:

[1] Hasan, Md Sazib and Krishnamoorthy, K (2018). Confidence intervals for the mean and a percentile based on zero-inflated lognormal data. Journal of Statistical Computation and Simulation, 88(8), 1499-1514.

[2] Li, Xinmin and Zhou, Xiaohua and Tian, Lili (2013). Interval estimation for the mean of lognormal data with excess zeros. Statistics & Probability Letters, 83(11), 2447-2453.

[3] Donner, Allan and Zou, Guang Yong (2011). Estimating simultaneous confidence intervals for multiple contrasts of proportions by the method of variance estimates recovery, Statistics in Biopharmaceutical Research, 3(2), 320-335.

[4] Harvey, J and Van der Merwe, AJ (2012). Bayesian confidence intervals for means and variances of lognormal and bivariate lognormal distributions. Journal of Statistical Planning and Inference, 142(6), 1294-1309. 

4. The authors must review the English. 

Response: Thank you for your suggestion. The manuscript has been carefully checked by a native English speaker before resubmitting this revision.

Materials and methods

1. Line 57, the authors let Y=log⁡W=(Y_j1,Y_j2…,Y_(jn_j(1) ))~N(μ,σ^2). However, Y is a vector random variable so if one assumes independence than one can assume Y∼N(μ,σ^2 I)., where I the identity matrix of dimension n_j(1) .

Response: Thank you for pointing this out. In this revised manuscript, we agree with your comments and so have changed it to Y∼N(μ,σ^2 I) in line 70.

2. Formula (3), what do the authors mean by Var(θ_i)? Unless you assume a Bayesian framework and so θj is a random variable this doesn’t make sense. Perhaps the authors intend Var(θ ^_j)? A similar mistake is repeated in lines 72-73. Even if the authors write Var(θ ^_ij) the equality would not hold unless you assume 〖COV(θ〗_i,θ_j)=0.

Response: We thank the reviewer for these helpful suggestions. In Eq (3), we intended using Var(θ ^_j), and so have done so throughout this manuscript. Moreover, in lines 86-88, we assume that θ ^_j and θ ^_l are independent random variables, and so the covariance becomes 〖COV(θ〗_i,θ_j)=0.

3. Formula (6) is an approximation of the estimated variance and not equality as reported.

Response: Thank you for this constructive suggestion. Eq (6) has been edited in this revision.

4. Formulae (7) and (8) are not correct.

Response: Eqs (7) and (8) have been edited, and we have added the details are related to both of equations, as shown in Lines 98-99.

5. Formula (13) would imply no difference between a simultaneous confidence interval derived for two or more than two groups. The authors should clarify this point. This fact also seems true for the MOVER interval.

Response: Thank you for this excellent suggestion. In Lines 110-112, we have offered an explanation of why there is no difference between them (Eq 13). 

6. I don’t understand whether Theorem (1) is proved for w_j>0, which would exclude the delta part of the log-normal distribution.

Response: Thank you for pointing this out. In Theorem 1 (S1 Appendix), we assume that the delta part is included in the information for θ_jl, which is the parameter of interest in this study.

7. Formula (22). This quantity was introduced in formula (9). The authors call the same quantity in several ways.

Response: Thank you for this comment. In Eq (15), we used a beta distribution with an FGPQ of δ^', while for Eq (9), random samples are drawn from the sampling distribution of δ^' (a beta distribution). Thus, a beta distribution is assumed in both but for different reasons.

8. Formula (28). Here the authors introduce the logarithm of the difference between two means. Then, in formula (29) they work on the difference between the means. Please be consistent.

Response: We thank the reviewer for this important criticism. The MOVER interval is suitable for the formulated CI for the combination of the parameters, so the delta-lognormal mean (θ_j) is log-transformed as β_j=ln⁡〖θ_j=ln⁡〖δ_j^'+(μ_j+σ_j^2)〗 〗 in Eq (20). After that, we can construct the MOVER interval for θ_j=exp⁡(β_j) in Eq (24). Finally, the MOVER-based SCI for θ_jl is obtained using Eq (25).

Simulation Study

The simulation study should involve the following measures

• Coverage probability (CP),

• Upper error and lower error,

• Relative length (RL),

as defined for example in [2]. The authors considered the CP and RL. However, the definition of these two measures is not the standard one (Lines 276-278).

Response: We thank the reviewer for pointing this out. The lower and upper error probabilities were added as measures for the SCI performances in the Simulation studies, as can be seen in Tables 1- 2.

Reference:

[2] Li, Xinmin and Zhou, Xiaohua and Tian, Lili (2013). Interval estimation for the mean of lognormal data with excess zeros. Statistics & Probability Letters, 83(11), 2447-2453.

An Example using Real Data

The dataset seems not respect one important assumption, i.e. independence between observation within the group

Response: We would like to thank the reviewer for pointing this important aspect out. Comparing the natural rainfall datasets for the five areas of Thailand is of interest in this study. It is possible that the observed rainfall datasets for the areas are independent due to their different climate patterns and meteorological conditions. Thailand is further sub-divided into the provinces, and there are meteorological stations in each of them. For example, in the northern area, the Chiang Mai and Uttaradit stations are located in topographical features that are conspicuously different: Chiang Mai is a city in a mountainous area while Uttaradit is in the Nan River valley with the Queen Sirikit Dam toward the north of the city. Thus, the observed rainfall amounts within the areas are independent as well.

We look forward to hearing from you in the near future.

Sincerely yours,

Patcharee Maneerat, Sa-Aat Niwitpong, and Suparat Niwitpong

The authors

---

## [Decision Letter · Decision Letter 1]

4 May 2021

PONE-D-20-38684R1

Simultaneous confidence intervals for all pairwise comparisons of the means of delta-lognormal distributions with application to rainfall data

PLOS ONE

Dear Dr. Niwitpong,

Thank you for submitting your manuscript to PLOS ONE. After careful consideration, we feel that it has merit but does not fully meet PLOS ONE’s publication criteria as it currently stands. Therefore, we invite you to submit a revised version of the manuscript that addresses the points raised during the review process.

We look forward to receiving your revised manuscript.

Kind regards,

Mohammad Asghari Jafarabadi

Academic Editor

PLOS ONE

Reviewers' comments:

Reviewer's Responses to Questions

**Comments to the Author**

1. If the authors have adequately addressed your comments raised in a previous round of review and you feel that this manuscript is now acceptable for publication, you may indicate that here to bypass the “Comments to the Author” section, enter your conflict of interest statement in the “Confidential to Editor” section, and submit your "Accept" recommendation.

Reviewer #2: All comments have been addressed

Reviewer #3: (No Response)

2. Is the manuscript technically sound, and do the data support the conclusions?

Reviewer #2: Yes

Reviewer #3: No

3. Has the statistical analysis been performed appropriately and rigorously? 

Reviewer #2: Yes

Reviewer #3: No

4. Have the authors made all data underlying the findings in their manuscript fully available?

Reviewer #2: Yes

Reviewer #3: Yes

5. Is the manuscript presented in an intelligible fashion and written in standard English?

Reviewer #2: Yes

Reviewer #3: Yes

6. Review Comments to the Author

Reviewer #2: Thank you for addressing my comments. The new manuscript is much more easily readable and better understandable. I recommend to accept this paper, I just have a few very minor recommendations:

1. I don't think it is necessary that each formula has its own number, I think it is enough to give a number to the ones which you further use or refer to. This makes the manuscript even easier to read. However, that's an editorial question and therefore it's not my decision if that should be changed or not.

2. Page 11, first sentence: Please delete "using RStudio". Instead, please add another line at the beginning of the results section: "R, version (...edit version...) was used for the computations of the simulations and the application." or something like that. Please note that it is not important if you use RStudio or not, it's the R version that counts.

3. Page 11, Simulation results, line 263: Delete "even".

4. Page 16, lines 288 - 290: This sentence is grammatically a bit weird and I didn't fully understand it. Please rewrite.

5. Page 17, line 291: Delete "the".

Reviewer #3: I am sorry, but I don't think the authors have adequately addressed my comments, and I feel that this manuscript is not acceptable for publication. Next, I will give some examples.

Example 1) I had raised the following point:

- The results look like a simple extension of the following papers [2], [1], [3] and [4]. The authors must clarify their contribution. For example, it is not clear whether the authors propose the theorems presented in the paper.

Authors response:

This is an excellent point raised by the reviewer. In Paragraph 4 of the Introduction, we have clarified our contribution and clarified its context in relation to studies [1–4]. In Supporting information (S1 Appendix), we have carefully checked the contexts of the theorems, as the reviewer requested.

I checked Paragraph 4 of the Introduction, where the authors write:

"Our contribution to the field is constructing SCIs based on our proposed methods to elucidate the pairwise differences between the means of multiple delta-lognormal 59 distributions."

which I don't think can be considered a satisfactory explanation.

Example 2) Technical mistakes. I had raised the following point:

- Formula (3), what do the authors mean by Var(θ_i)? Unless you assume a Bayesian framework and so θj is a random variable, this doesn't make sense. Perhaps the authors intend Var(θ ^_j)? A similar mistake is repeated in lines 72-73. Even if the authors write Var(θ ^_ij) the equality would not hold unless you assume 〖COV(θ〗 _i,θ_j)=0.

Authors response:

We thank the reviewer for these helpful suggestions. In Eq (3), we intended using Var(θ ^_j), and so have done so throughout this manuscript. Moreover, in lines 86- 88, we assume that θ ^_j and θ ^_l are independent random variables, and so the covariance becomes 〖COV(θ〗_i,θ_j)=0.

So, the authors correct the first technical mistake. However, it doesn't make sense to assume θ ^_j and θ ^_l as independent random variables since these quantities are estimated from the data, and so the dependence must be proved.

Example 3) Technical mistakes. I had raised the following point:

-Formulae (7) and (8) are not correct.

Response: Eqs (7) and (8) have been edited, and we have added the details are related to both of equations, as shown in Lines 98-99.

Unfortunately, Formulae (7) and (8) are still not correct.

7. PLOS authors have the option to publish the peer review history of their article (what does this mean?). If published, this will include your full peer review and any attached files.

Reviewer #2: **Yes: **Julia Braun

Reviewer #3: **Yes: **Antonino Abbruzzo

---

## [Author Response · Author response to Decision Letter 1]

31 May 2021

Response to Reviewers

Journal: PLOS One

Manuscript ID: PONE-D-20-38684R1

Title name: Simultaneous confidence intervals for all pairwise comparisons of the means of delta-lognormal distributions with application to rainfall data

Authors: Patcharee Maneerat, Sa-Aat Niwitpong and Suparat Niwitpong

Dear Reviewers: 

We would like to thank you for your time as well as patience to go over manuscript and making constructive suggestions to improve it for a resubmission. To address the specific points raised by the reviewers, the following responses are to be noted:

 Reviewer #2: 

1. I don't think it is necessary that each formula has its own number, I think it is enough to give a number to the ones which you further use or refer to. This makes the manuscript even easier to read. However, that's an editorial question and therefore it's not my decision if that should be changed or not.

Response: We thank the reviewer for these helpful suggestions. We agree with your comments and so have given an equation number that was referred in this revised manuscript especially.

2. Page 11, first sentence: Please delete "using RStudio". Instead, please add another line at the beginning of the results section: "R, version (...edit version...) was used for the computations of the simulations and the application." or something like that. Please note that it is not important if you use RStudio or not, it's the R version that counts.

Response: Thank you for pointing this out. This point was moved to the beginning of the results section.

3. Page 11, Simulation results, line 263: Delete "even".

Response: Line 276, it was deleted.

4. Page 16, lines 288 - 290: This sentence is grammatically a bit weird and I didn't fully understand it. Please rewrite.

Response: Thank you for pointing this out. We have rewritten in lines 301-304.

5. Page 17, line 291: Delete "the".

Response: Line 305, it was deleted.

 Reviewer #3: 

I am sorry, but I don't think the authors have adequately addressed my comments, and I feel that this manuscript is not acceptable for publication. Next, I will give some examples.

1. Example 1) I had raised the following point:

- The results look like a simple extension of the following papers [2], [1], [3] and [4]. The authors must clarify their contribution. For example, it is not clear whether the authors propose the theorems presented in the paper. 

Authors response:

This is an excellent point raised by the reviewer. In Paragraph 4 of the Introduction, we have clarified our contribution and clarified its context in relation to studies [1–4]. In Supporting information (S1 Appendix), we have carefully checked the contexts of the theorems, as the reviewer requested.

Response: Paragraph 3 of the Introduction, we have clarified in relation to studies [1–4]. Motivated by studies [1–4] and [5-8], the proofs for the PB, FGCI and MOVER-based SCI with the asymptotic coverage property are modified for the present study, as mentioned after Eqs. (5), (10) and (13), respectively.

References:

[1] Hasan, Md Sazib and Krishnamoorthy, K (2018). Confidence intervals for the mean and a 

percentile based on zero-inflated lognormal data. Journal of Statistical Computation and Simulation, 88(8), 1499-1514.

[2] Li, Xinmin and Zhou, Xiaohua and Tian, Lili (2013). Interval estimation for the mean of lognormal data with excess zeros. Statistics & Probability Letters, 83(11), 2447-2453.

[3] Donner, Allan and Zou, Guang Yong (2011). Estimating simultaneous confidence intervals for multiple contrasts of proportions by the method of variance estimates recovery, Statistics in Biopharmaceutical Research, 3(2), 320-335.

[4] Harvey, J and Van der Merwe, AJ (2012). Bayesian confidence intervals for means and variances of lognormal and bivariate lognormal distributions. Journal of Statistical Planning and Inference, 142(6), 1294-1309. 

[5] Hannig, J., Abdel-Karim, L. E, A. and Iyer, H. K. (2006) Simultaneous Fiducial Generalized Confidence Intervals for Ratios of Means of Lognormal Distributions, Austrian Journal of Statistics, 35(2-3), 261-269.

[6] Kharrati-Kopaei M, Eftekhar S. (2017). Simultaneous Confidence Intervals for Comparing Several Inverse Gaussian Means under Heteroscedasticity. Journal of Statistical Computation and Simulation, 87(4), 777-790.

[7] Li J, Song W, Shi J. (2015). Parametric Bootstrap Simultaneous Confidence Intervals for Differences of Means from Several Two-Parameter Exponential Distributions. Statistics & Probability Letters, 106, 39-45.

[8] Thangjai W, Niwitpong S. (2020). Simultaneous Confidence Intervals for All Differences

of Coefficients of Variation of Two-Parameter Exponential Distributions. Thailand Statistician, 18(2), 135-149.

2. I checked Paragraph 4 of the Introduction, where the authors write:

"Our contribution to the field is constructing SCIs based on our proposed methods to elucidate the pairwise differences between the means of multiple delta-lognormal distributions."

which I don't think can be considered a satisfactory explanation.

Response: Paragraph 4 of the Introduction, we have revised our contribution as well as the motivation behind this study which is the estimated pairwise differences between the mean of natural rainfall records from the five areas of Thailand. 

3. Example 2) Technical mistakes. I had raised the following point:

- Formula (3), what do the authors mean by Var(θ_i)? Unless you assume a Bayesian framework and so θj is a random variable, this doesn't make sense. Perhaps the authors intend Var(θ ^_j)? A similar mistake is repeated in lines 72-73. Even if the authors write Var(θ ^_ij) the equality would not hold unless you assume 〖COV(θ〗 _i,θ_j)=0.

Authors response:

We thank the reviewer for these helpful suggestions. In Eq (3), we intended using Var(θ ^_j), and so have done so throughout this manuscript. Moreover, in lines 86- 88, we assume that θ ^_j and θ ^_l are independent random variables, and so the covariance becomes 〖COV(θ〗_i,θ_j)=0.

So, the authors correct the first technical mistake. However, it doesn't make sense to assume θ ^_j and θ ^_l as independent random variables since these quantities are estimated from the data, and so the dependence must be proved.

Response: Lines 88-92, this point has been clarified the reason of the covariance becoming 〖COV(θ ^〗_i,θ ^_j)=0 , as shown in Crow and Shimizu [21].

Reference:

[21] Crow EL, Shimizu K. Lognormal Distributions: Theory and Applications. New York, N.Y.; Basel: CRC Press; 1988.

4. Example 3) Technical mistakes. I had raised the following point:

-Formulae (7) and (8) are not correct.

Response: Eqs (7) and (8) have been edited, and we have added the details are related to both of equations, as shown in Lines 98-99. Unfortunately, Formulae (7) and (8) are still not correct.

Response: For the PB interval, the sizes of positive observations are not equal for each resampling with replacement under bootstrap approach, so the Eqs (2) and (3) have been edited. 

We look forward to hearing from you in the near future.

Sincerely yours,

Patcharee Maneerat, Sa-Aat Niwitpong and Suparat Niwitpong

The authors

---

## [Decision Letter · Decision Letter 2]

16 Jun 2021

Simultaneous confidence intervals for all pairwise comparisons of the means of delta-lognormal distributions with application to rainfall data

PONE-D-20-38684R2

Dear Dr. Niwitpong,

We’re pleased to inform you that your manuscript has been judged scientifically suitable for publication and will be formally accepted for publication once it meets all outstanding technical requirements.

Kind regards,

Mohammad Asghari Jafarabadi

Academic Editor

PLOS ONE

Reviewers' comments:

Reviewer's Responses to Questions

**Comments to the Author**

1. If the authors have adequately addressed your comments raised in a previous round of review and you feel that this manuscript is now acceptable for publication, you may indicate that here to bypass the “Comments to the Author” section, enter your conflict of interest statement in the “Confidential to Editor” section, and submit your "Accept" recommendation.

Reviewer #2: All comments have been addressed

2. Is the manuscript technically sound, and do the data support the conclusions?

Reviewer #2: Yes

3. Has the statistical analysis been performed appropriately and rigorously? 

Reviewer #2: Yes

4. Have the authors made all data underlying the findings in their manuscript fully available?

Reviewer #2: Yes

5. Is the manuscript presented in an intelligible fashion and written in standard English?

Reviewer #2: Yes

6. Review Comments to the Author

Reviewer #2: (No Response)

7. PLOS authors have the option to publish the peer review history of their article (what does this mean?). If published, this will include your full peer review and any attached files.

Reviewer #2: **Yes: **Julia Braun

---

## [Editor Report · Acceptance letter]

24 Jun 2021

PONE-D-20-38684R2 

Simultaneous confidence intervals for all pairwise comparisons of the means of delta-lognormal distributions with application to rainfall data 

Dear Dr. Niwitpong:

I'm pleased to inform you that your manuscript has been deemed suitable for publication in PLOS ONE. Congratulations! Your manuscript is now with our production department. 

Kind regards, 

on behalf of

Professor Mohammad Asghari Jafarabadi 

Academic Editor

PLOS ONE